# Application of Optimization Techniques for Searching Optimal Reservoir Rule Curves: A Review

**Anongrit Kangrang** [1],*[ID]**, Haris Prasanchum** [2]**, Krit Sriworamas** [3]**, Seyed Mohammad Ashrafi** [4][ID]**, Rattana Hormwichian** [1]**, Rapeepat Techarungruengsakul** [1] **and Ratsuda Ngamsert** [1]

1   Faculty of Engineering, Mahasarakham University, Kantarawichai, Maha Sarakham 44150, Thailand
2   Faculty of Engineering, Rajamangala University of Technology Isan, Khon Kaen Campus, Khon Kaen 40000, Thailand
3   Faculty of Engineering, Ubon Ratchathani University, Ubon Ratchathani 34190, Thailand
4   Department of Civil Engineering, Faculty of Civil Engineering and Architecture, Shahid Chamran University of Ahvaz, Ahvaz 83151-61355, Iran
*   Correspondence: anongrit.k@msu.ac.th

**Abstract:** This paper reviews applications of optimization techniques connected with reservoir simulation models to search for optimal rule curves. The literature reporting the search for suitable reservoir rule curves is discussed and examined. The development of optimization techniques for searching processes are investigated by focusing on fitness function and constraints. There are five groups of optimization algorithms that have been applied to find the optimal reservoir rule curves: the trial and error technique with the reservoir simulation model, dynamic programing, heuristic algorithm, swarm algorithm, and evolutionary algorithm. The application of an optimization algorithm with the considered reservoirs is presented by focusing on its efficiency to alleviate downstream flood reduction and drought mitigation, which can be explored by researchers in wider studies. Finally, the appropriate future rule curves that are useful for future conditions are presented by focusing on climate and land use changes as well as the participation of stakeholders. In conclusion, this paper presents the suitable conditions for applying optimization techniques to search for optimal reservoir rule curves to be effectively applied in future reservoir operations.

**Keywords:** optimal reservoir rule curves; optimization algorithm; climate change; reservoir model; stakeholder participation

## 1. Introduction

Despite numerous advances in watershed management, there are still several factors that lead to extreme events, such as floods and droughts, in various regions, particularly Southeast Asia and Thailand. Some of the main factors include global warming, limited natural areas, increased industrial usage and agricultural land development, which can significantly impact hydrological processes in basins [1–3]. To achieve a balance between resources and demand, water resource management adopts non-structural solutions. The management of the supply side requires improved river–reservoir system operations, with reservoir rule curves acting as essential guidelines for regulating the amount of water released at different times [4,5]. Therefore, modifying the reservoir rule curves to consider the effects of land use and climate change is necessary.

Typically, reservoir rule curves are derived from the optimization of the system operation process using historical hydrological data and physical system constraints. To manage long periods, including drought and flood events, simulation models evaluate the effectiveness of the achieved rules and modify them if needed. In Thailand, the operating rule curves of large reservoirs are revised every five years using updated hydrological time series [6,7]. However, the inflow to the reservoir system undergoes significant changes due to climate and land use alterations, making it impossible to accurately predict future

operations using past hydrological data alone [8,9]. Therefore, implementing a mechanism to predict hydrological flows entering the system is necessary when modifying reservoir operating rule curves. The new rule curves, extracted using accurate forecast models that consider the impacts of climate change and land use development, will likely perform better in dealing with future hydrological events. In general, reviewing operating policies within a technical framework that includes interdisciplinary approaches and stakeholder participation is crucial.

To achieve its efficient management, reservoir operation employs a combination of optimization techniques, the prediction of future hydrological conditions, and the participation of experienced decision-makers, with cross-disciplinary and stakeholder approaches playing central roles. The participation of stakeholders promotes an optimization process that incorporates the experience of reservoir operators [10]. Additionally, the quality of the obtained results largely depends on the applied optimization algorithm [11]. Therefore, this study presents and reviews some conventional optimization algorithms from the literature for comparative purposes. The contribution of this paper includes presenting the suitable conditions to apply optimization techniques alongside a reservoir simulation model to search for the accepted rule curves, including fitness functions, searching constraints, land use change, climate change, and the participation of stakeholders. Details of each contribution are presented within the paper.

## 2. Reservoir Simulation Models

Reservoir simulation models were developed to represent physical phenomena under specific conditions, utilizing the principles of mass conservation and the physical limitations of the reservoir and its associated facilities through interconnected mathematical relationships [12–15]. These models are useful tools for system managers to estimate the outcomes of their decisions, enabling them to achieve their goals and produce the highest economic returns. However, the accuracy of these models in estimating reservoir efficiency is heavily dependent on the correct estimation of reservoir inflows, a hydrological factor with high uncertainty [16].

To ensure the safe capacity of the reservoir, the solid yield is considered as the bare minimum and is transformed into the maximum permissible amount that can be utilized during critical times. This approach guarantees that enough water is available in the reservoir to meet the basic water needs of the downstream region. The first step in using rule curves to calculate a reservoir's water balance is to determine its storage capacity, starting at full capacity. Water release can then be calculated based on predefined policies, such as the standard operating policy, hedging rule, hydropower rule, and space rule [17–19]. SOP have been developed and widely applied because of the need to meet target demands. Therefore, they are suitable for using to control water release for irrigation, water supply and power generation. On the other hand, HR have been developed for the operation of reservoirs during dry seasons under different conditions, serving the high fluctuations of inflow. HR release criteria include allocating irrigation water considering the vast effects of climate change, whilst focusing on mitigating current and future droughts. Moreover, HR release criteria can be used in conjunction with reservoir rule curves for managing both flood and drought situations [20]. Figure 1 presents a conceptual diagram of the hedging rule (HR) and standard operating policy (SOP), which can be expressed mathematically by Equations (1) and (2), respectively.

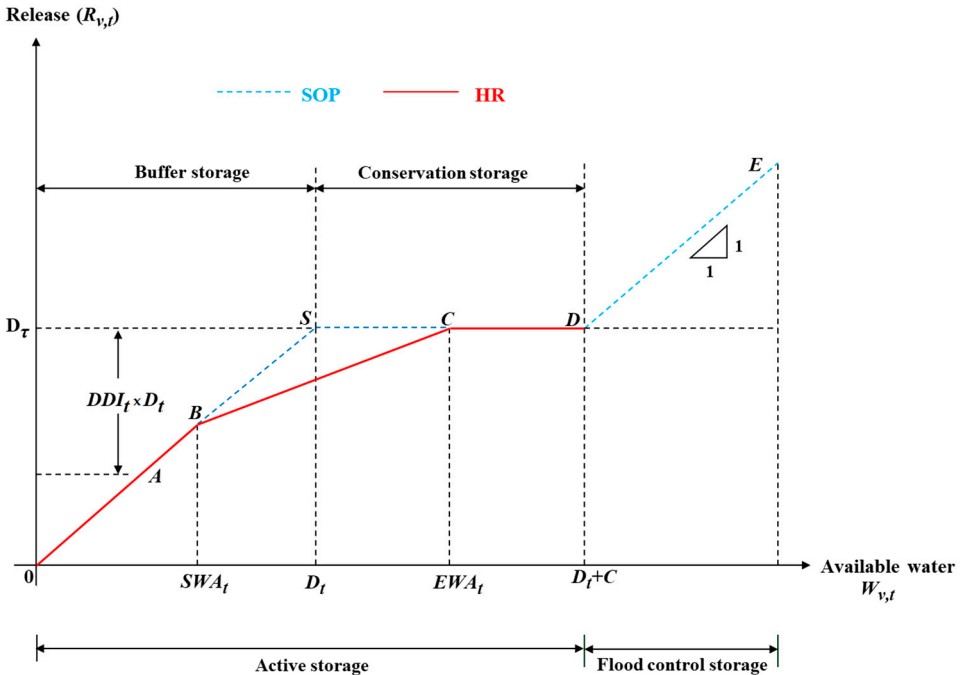

**Figure 1.** The conceptual diagram of HR and SOP.

The hedging policy is expressed as follows:

$$\text{When } 0 \leq (1 - DDIt) \cdot Dt \leq SWAt$$

$$R_{v,\tau} = \begin{cases} WA_\tau & \text{if } WA_\tau < SWA_\tau \\ D_\tau + (SWA_\tau - D_\tau)\frac{WA_\tau - EWA_\tau}{SWAt_\tau - EWA_\tau} & \text{if } SWA_\tau \leq WA_\tau \leq EWA_\tau \\ D_\tau & \text{if } EWA_\tau \leq WA_\tau < DD_\emptyset + C \\ WA_\tau - C & \text{if } WA_\tau \geq D_\emptyset + C \\ 0, \text{otherwise} \end{cases} \quad (1)$$

where $R_{v,\tau}$ is the discharge of the reservoir at time $\tau$; $SWA_\tau$ and $EWA_\tau$ represent the available water at the start and end points at time $\tau$, respectively; and $D_\tau$ is the downstream water demand at time $\tau$.

The standard operating policy is expressed as follows:

$$R_{v,\tau} = \begin{cases} D_\tau + W_{v,\tau} - D_\tau + C, for \ W_{v,\tau} \geq D_\tau + C + D_\tau \\ D\tau, for \ D_\tau \leq W_{v,\tau} < D_\tau + C + D_\tau \\ D_\tau + W_{v,\tau} - D_v \ for \ D_\tau - D_\tau \leq W_{v,\tau} < D_\tau \\ 0, otherwise \end{cases} \quad (2)$$

where $R_{v,\tau}$ is the discharge of water during year ν and month $\tau$ ($\tau$ = 1 to 12, standing for January to December); $D_\tau$ is the net downstream water demand during month $\tau$; $D_t$ is the lower rule curve of month $\tau$; $D_t + C$ is the upper rule curve of month $\tau$; and $W_{v,\tau}$ is the available water during year ν and month $\tau$, as described in Equation (3):

$$W_{v,\tau} = S_{v,\tau-1} + Q_{v,\tau} - R_{v,\tau} - E_\tau \quad (3)$$

where $S_{v,\tau-1}$ is the stored water at the end of month $\tau - 1$; $Q_{v,\tau}$ is the monthly inflow to the reservoir; and $E_\tau$ is the evaporation loss.

The performance criteria of the system, including the number of failure periods, the amount of supply and demand, the amount of excess water (spill), and the maximum and average vulnerabilities, can be calculated as the results of the simulation model. Figure 2

shows the regulatory water released from a reservoir and its characteristics compared to target demands.

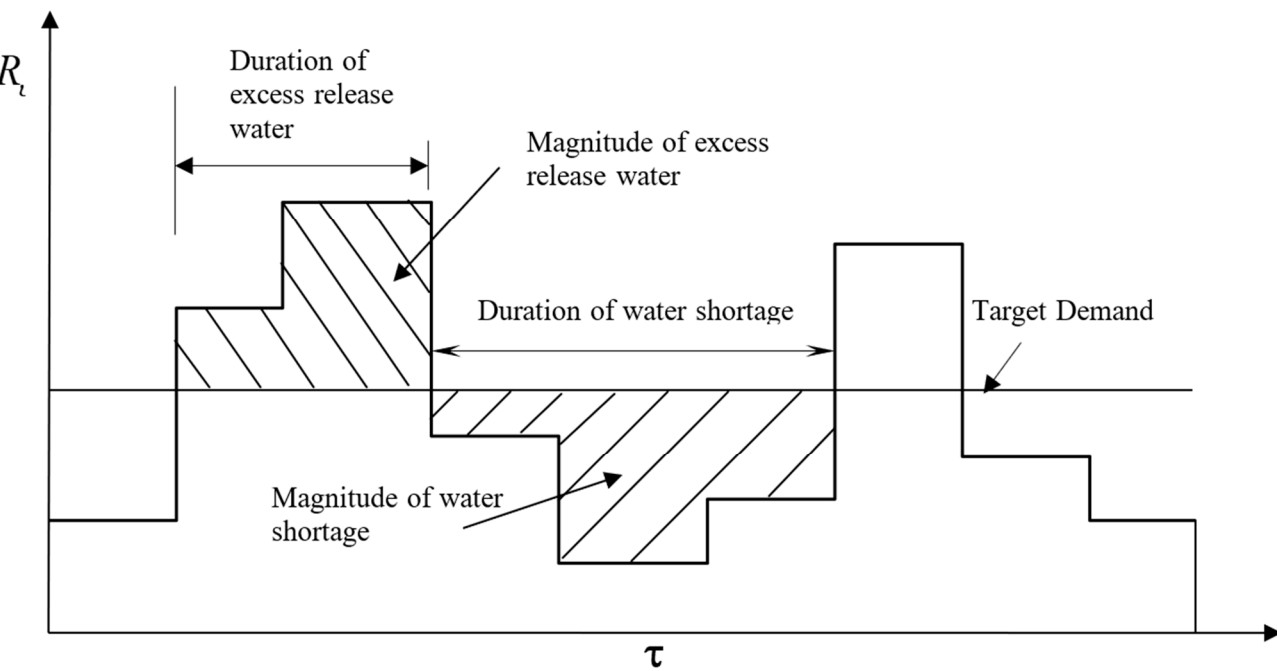

**Figure 2.** Characteristics of water shortage, excess release water, and target demands.

## 3. Optimization Techniques for Reservoir Rule Curve Extraction

The optimization–simulation technique is a popular approach for the optimization and system simulation of water resources. This technique uses optimization models to find the best solutions while also simulating the system details. By connecting optimization techniques with the reservoir model using monthly rule curves as decision variables, the best values for these variables can be found to provide the best fitness for the objective function. Optimizing the key points of rule curves can be an effective manner of avoiding their fluctuation. Therefore, this section is divided into three sections, including the integration of optimization techniques and the reservoir simulation model, the searching process for the objective function, and optimizing the points of the obtained rule curves.

### 3.1. Integrating Optimization Techniques and the Reservoir Simulation Model

Reservoir rule curves consist of upper and lower curves, with 24 decision variables in total for one reservoir. The physical characteristics of the reservoir and operational constraints determine the feasible region, or the boundary, for each variable, which typically ranges from dead storage to full capacity. A visualization of the optimization–simulation technique, including the integration of optimization techniques and the reservoir model, and the boundary of decision variables, is presented in Figures 3 and 4. Once the optimal rule curves are found, they are used as the release criteria in the reservoir simulation model, which considers both historic and synthetic inflows. This evaluation process results in the creation of management criteria, such as the frequency, intensity, amount, and duration of flood or drought conditions, as shown in Figure 5. In summary, the optimization–simulation technique is an effective tool for optimizing water resource management strategies. The optimization techniques can be integrated with the reservoir simulation model, both as a single or multi-reservoir system. The decision variables (rule curves value) for a single reservoir are 24 values, with 12 values for the lower rule curves and 12 values for the upper rule curves. On the other hand, the multi-reservoir system uses 48 values for two reservoirs, 72 for three reservoirs, and 96 for four reservoirs. The search process simultaneously provides the optimal rule curves of all the reservoirs of the system [21].

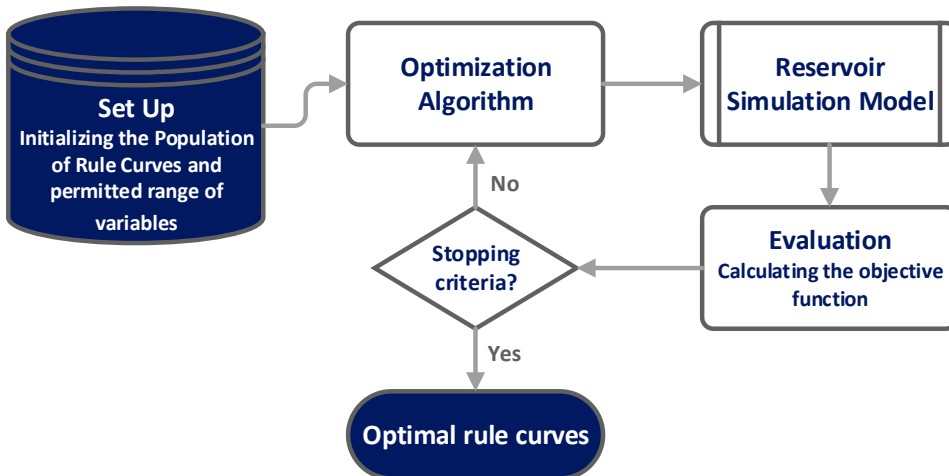

**Figure 3.** Heuristic integration of the optimization algorithm and reservoir simulation model to search for the rule curves.

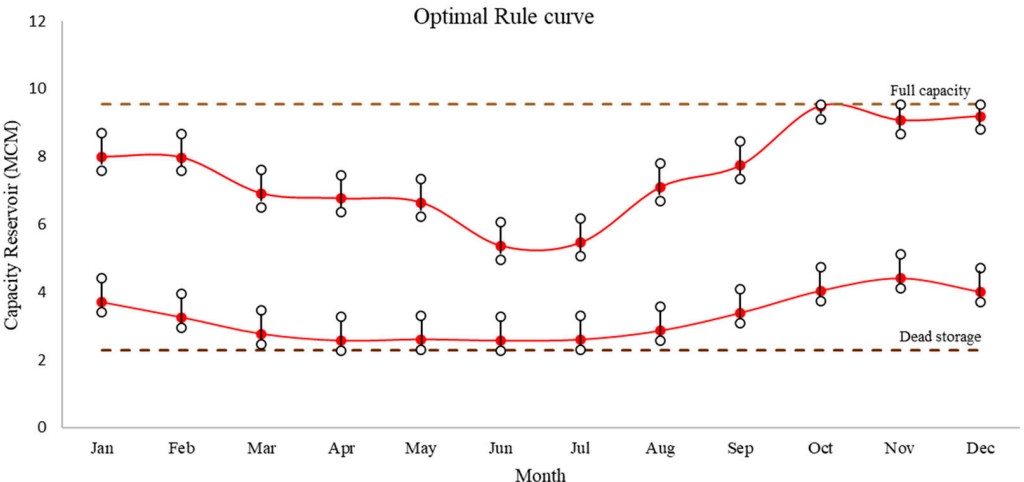

**Figure 4.** Boundary of the search process for the optimal rule curves.

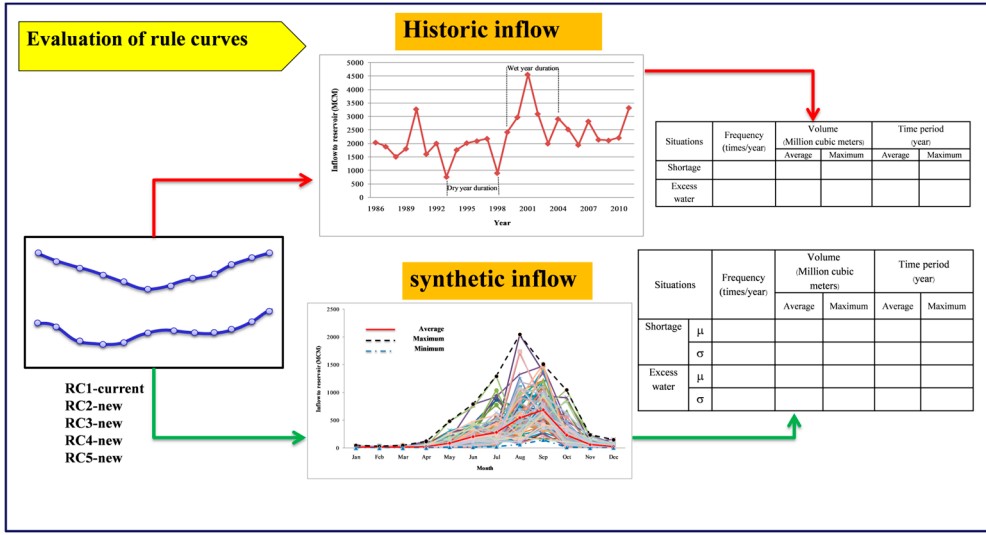

**Figure 5.** Evaluation process for the optimal rule curves.

### 3.2. Objective Function of the Search Process

Generally, most objective functions that search for an optimal solution are calculated from the results of a simulation model. In our case, these values represent flood and drought situations up to the limits imposed by the physical characteristics of each reservoir. The popular objective functions for searching the optimal reservoir rule curves are the minimal average water shortage described in Equation (4), the minimal frequency of water shortages in Equation (5), the minimal average excess water per year in Equation (6), and the minimal frequency of excess water in Equation (7) [8,9], shown as follows:

The minimal average water shortage per year:

$$MinH_{(avr)} = \frac{1}{n} \sum_{v=1}^{n} Sh_V \tag{4}$$

The minimal frequency of water shortage:

$$MinFre_{(i)} = \frac{1}{n} \sum_{v=1}^{n} Y_{shv} \tag{5}$$

The minimal average excess water per year:

$$MinP_{(avr)} = \frac{1}{n} \sum_{v=1}^{n} Sp_V \tag{6}$$

The minimal frequency of excess water:

$$MinFre_{(i)} = \frac{1}{n} \sum_{v=1}^{n} Y_{spv} \tag{7}$$

where $H_{(avr)}$ is the average water shortage per year; $Fre_{(i)}$ is the frequency of water shortages; $P_{(avr)}$ is the average excess water per year; $n$ represents an entire year; $Sh_v$ is the water shortage in year $v$; $Y_{shv}$ is the year of water shortage; $Sp_v$ is the release of excess water during year $v$; and $Y_{spv}$ is the year of the release of excess water.

### 3.3. Optimizing the Points of the Rule Curves

The obtained rule curves often fluctuate and lack practical usefulness due to the independence of the searching boundary for each decision variable. These fluctuations are caused by the seasonal pattern of streamflow and make it challenging, if not impossible, to use the resulting curves. To avoid fluctuations in the obtained values, the smoothing function and the moving average constraint are incorporated with the constraints of the searching procedure. Firstly, a smoothing function can be applied to both the upper and lower rule curves. This function controls the maximum and minimum storage levels of the reservoir at the beginning and end of the flood season, respectively [22].

To describe the smoothing function for adjusting the rule curves, the monthly rule curves' lower and upper levels were fixed and represented by the 'x' and 'y' values, respectively. In the early drought season, the water level in January $(x_1, y_1)$ should be higher than that in February $(x_2, y_2)$ and then gradually decrease until June $(x_6, y_6)$. From July $(x_7, y_7)$, which marks the beginning of the flood season, the water level should be higher than that in June $(x_6, y_6)$, and in August $(x_8, y_8)$, the water level should rise higher than in July $(x_7, y_7)$ and increase until the end of the rainy season in October $(x_{10}, y_{10})$ until the end of December $(x_{12}, y_{12})$. This pattern reflects the seasonal streamflow of Thailand. The smoothing function constraints are integrated into the fitness function of the reservoir simulation model to fit the rule curves, as presented below:

$$x_1 > x_2 > x_3 > x_4 > x_5 > x_6 < x_7 < x_8 < x_9 < x_{10} < x_{11} < x_{12} \tag{8}$$

$$y_1 > y_2 > y_3 > y_4 > y_5 > y_6 < y_7 < y_8 < y_9 < y_{10} < y_{11} < y_{12} \tag{9}$$

where the variables $x_1$ and $y_1$ represent the initial water level in January, which marks the beginning of the drought season. Similarly, $x_5$ and $y_5$ represent the water level in May; $x_6$ and $y_6$ represent the initial water level in June, which marks the end of the drought season and the beginning of the flood season, and marks the starting point of the lower rule curve for the flood season. Lastly, $x_{12}$ and $y_{12}$ represent the water level in December, which marks the end of the upper rule curve for the flood season.

Secondly, the moving average is another smoothing function constraint for reducing the fluctuate rule curves, for each rule curve can be described as:

$$\left| \frac{x_{\tau-2} + x_{\tau-1} + x_\tau}{3} - x_\tau \right| \leq 0.1T \text{ for } \tau = 3, \ldots 12 \tag{10}$$

$$\left| \frac{x_{12} + x_{\tau-1} + x_\tau}{3} - x_\tau \right| \leq 0.1T \text{ for } \tau = 2 \tag{11}$$

$$\left| \frac{x_{12} + x_{12-\tau} + x_\tau}{3} - x_\tau \right| \leq 0.1T \text{ for } \tau = 1 \tag{12}$$

where $x$ is rule curve level and $T$ is the active storage of each reservoir. These smoothing functions are integrated into the fitness function of the searching procedure [22].

## 4. Typically Applied Optimization Techniques

The optimal rule curves have been improved by many methods, initially using easy methods and then more complicated procedures, such as the trial and error technique with the reservoir simulation model, dynamic programing, heuristic algorithm, swarm algorithm, and evolutionary algorithm. There are four groups of methods that are classified according to the solution criteria. Figure 6 shows the method groups that are usually used to find the optimal rule curves. Details of each technique are presented, starting with the simple techniques and followed by the more complicated methods.

### 4.1. Trial and Error Technique with the Reservoir Simulation Model

The trial and error technique is a basic method used to improve reservoir rule curves by using the reservoir simulation model [6,23]. This method involves evaluating the efficacy of the system using a number of trial rule curves, which are modified by individuals with previous experience in operating reservoirs. Figure 7 illustrates the procedure of the trial and error method for finding suitable rule curves. However, this method is not always acceptable in practice since it does not guarantee obtaining the best solution. Nevertheless, it can be considered as the basic version of the expert systems in the management of water resources.

### 4.2. Dynamic Programming

The dynamic programming algorithm (DP) is popular for solving combinatorial optimization problems [24–27]. Since the operating rules and performance characteristics of the reservoir system are usually nonlinear and naturally combinatorial, DP can be considered a very suitable method for optimizing the rule curves. It should be noted that dynamic programming suffers from the curse of dimensionality in large-scale problems. In previous research in this field, this issue has been pointed out as the greatest challenge of using DP in reservoir operation optimization. To tackle this problem, various modified versions of DP have been proposed in the literature, including discrete DP techniques [28,29], increment DP [30], discrete differential DP [31], incremental DP successive approximation [32], and folded DP [33]. Additionally, it should be noted that, due to the continuous nature of the problem variables (e.g., storage levels, release values, and rule curve levels), discrete algorithms are not suitable to solve reservoir management problems.

To overcome the above-mentioned dimensional problem, the principle of progressive optimality (PPO), a computationally efficient technique for a continuous state variable, is usually applied to find the optimal rule curves. It has been shown that the computational dimensionality of the DP/PPO approach is smaller than that of the traditional DP one (see Figure 8). The DP/PPO was applied to find the optimal rule curves for single and multi-reservoir systems in Thailand [34].

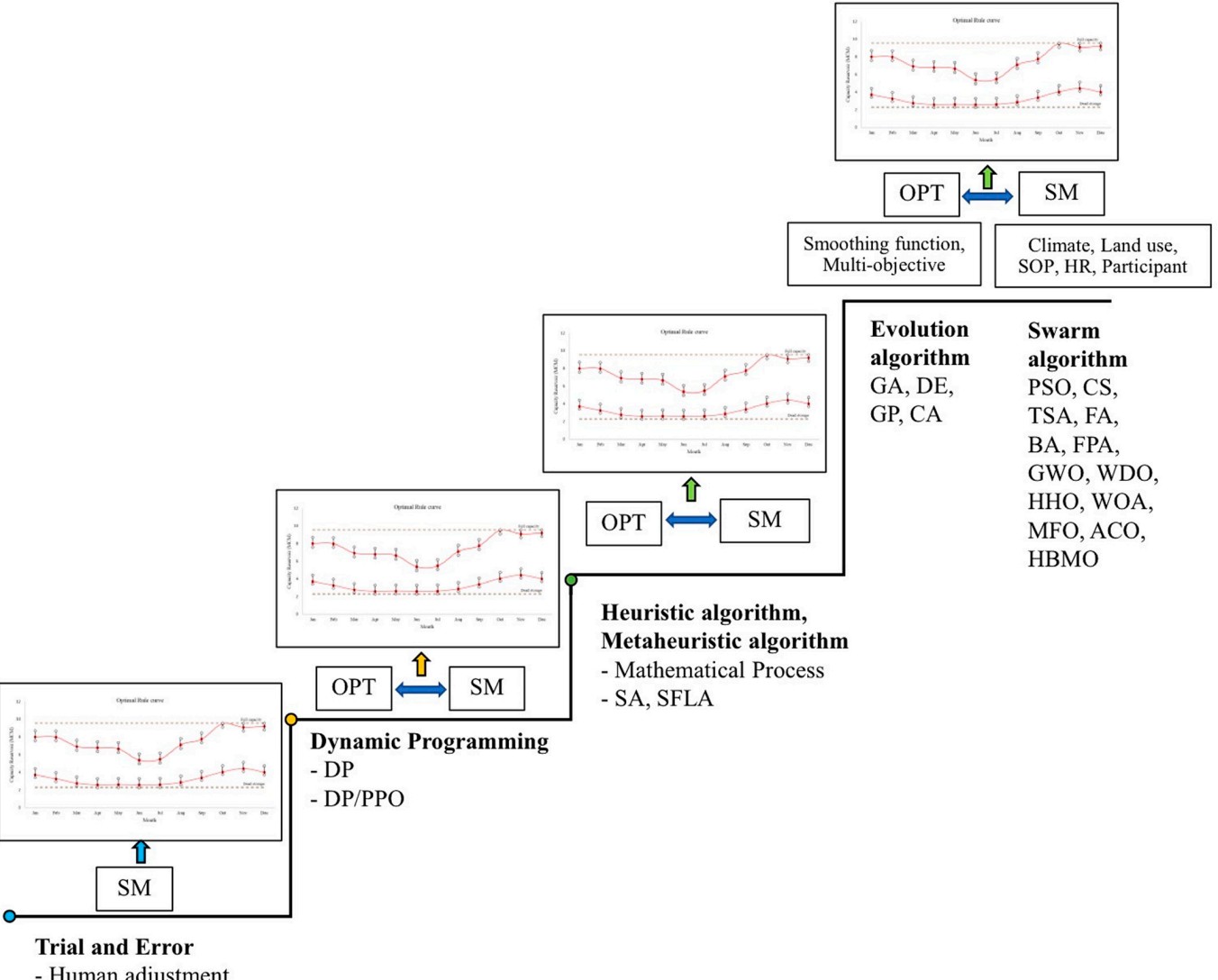

**Figure 6.** Groups of methods for finding the optimal rule curves.

*4.3. Heuristic and Metaheuristic Algorithms*

Heuristic algorithms (Has) use simple techniques for a guided random search to find the optimal solution. Some of them are based on a local search. Conceptual simplicity and ease of implementation have made such methods very popular. In this method, the search process starts with a random initial solution. The initial solution is modified by a simple operator, such as mutation. If this modification leads to the achievement of a more optimal solution, the new solution replaces the previous one. Otherwise, the current solution is retained. The processes of modification and replacement of the better solution continue iteratively until the stopping condition is satisfied. After the calculations are stopped, the best solution that is found is presented as the obtained optimum [35–37].

Recently, various Has have been applied to achieve the optimum operating policies for water resources [38–42].

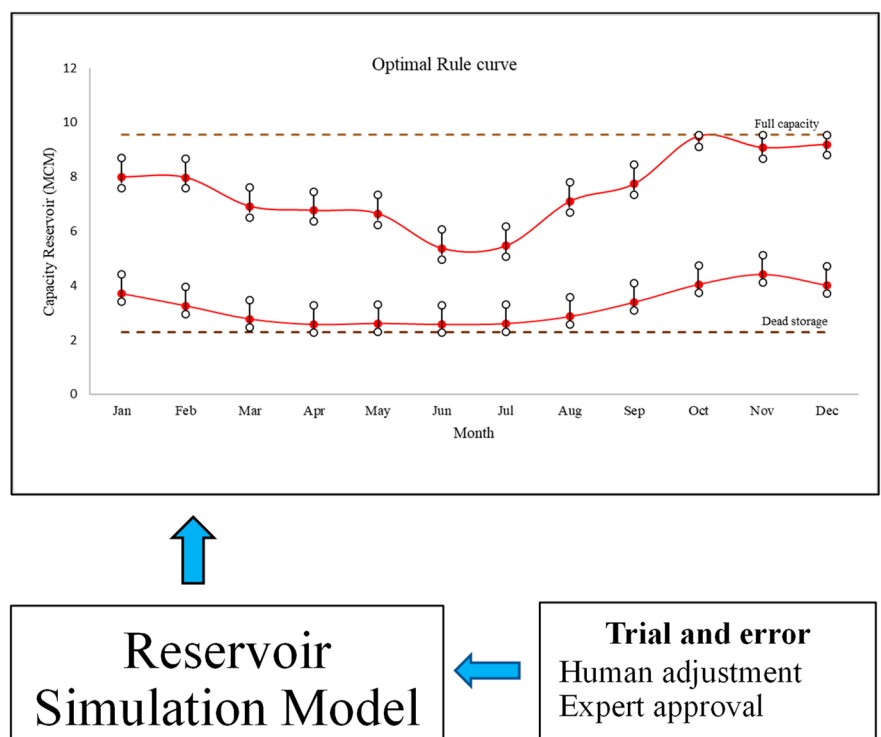

**Figure 7.** Procedure of the trial and error method for finding suitable rule curves.

HAs have been applied with the simulation model search to obtain the optimal reservoir rule curves. The heuristic algorithm begins with one initial solution to the problem, usually chosen at random (lower and upper rule curves). Those initial solutions are used in the reservoir simulation model. Then, the water discharge is calculated in each month based on the random rule curves for controlling water storage as an objective function [43] (see Figure 9). HAs have been applied to search optimal rule curves in multipurpose reservoirs in Thailand. The results showed that the reservoir rule curves developed with the HA technique were able to reduce water shortages by 43–44% compared with the existing rule curves [43]. In addition, HAs have also been applied to control the water allocation for the downstream reservoirs of the upper Tone River in Japan to prevent flooding during heavy rainfall using the objective function to minimize the difference between the simulated discharge thresholds and the actual operating criteria. It was found that, during the operation, flood peaks were effectively reduced [44].

Some of the heuristic and metaheuristic algorithms that have been integrated with the reservoir simulation model for searching optimal rule curves include the simulated annealing algorithm (SA) and shuffled frog leaping algorithm (SFLA).

### 4.3.1. Simulated Annealing Algorithm

The simulated annealing algorithm (SA) has been successfully utilized to solve optimization problems [45,46]. This approach is particularly effective in cases where the search space is discrete, and exhaustive enumeration is not practical due to limited time. Although SA may not always find the most efficient solution, it has proven to be an excellent strategy for solving organizational optimization problems [47]. In the field of reservoir management, SA has been extensively studied and applied in conjunction with the '10 reservoir problem' standard criteria [48]. SA has been employed in many real reservoir systems in Thailand to derive the optimal operating policies, where it has been found to take less computation time compared to other optimization techniques, such as the genetic algorithm (GA) [49]. For

example, SA has been applied to develop rule curves for the Bhumibol and the Sirikit Reservoirs in northern Thailand, which are large multipurpose reservoirs with an integrated management system. The resulting rule curves were effective in reservoir management, especially for seasonally fluctuating hydrological conditions [50]. SA has also been used to develop a non-linear time-dependent dynamic model to describe the operation of a single-purpose reservoir during the irrigation season. The objective of this model was to maximize the total farm income by optimizing the allocation of irrigated crops, while considering the changing conditions of soil moisture and plant water demand with an integrated water balance model. This model has been used as a decision support tool for irrigated cropping patterns and irrigation scheduling, such as in the planned reservoir on the Havrias River in northern Greece [51]. Currently, SA is still popular to develop optimal reservoir management and operational criteria, especially in integrating ecosystem-related data with water allocation to support the environmental and agricultural water demand and hydropower operation of multi-reservoir systems [52,53].

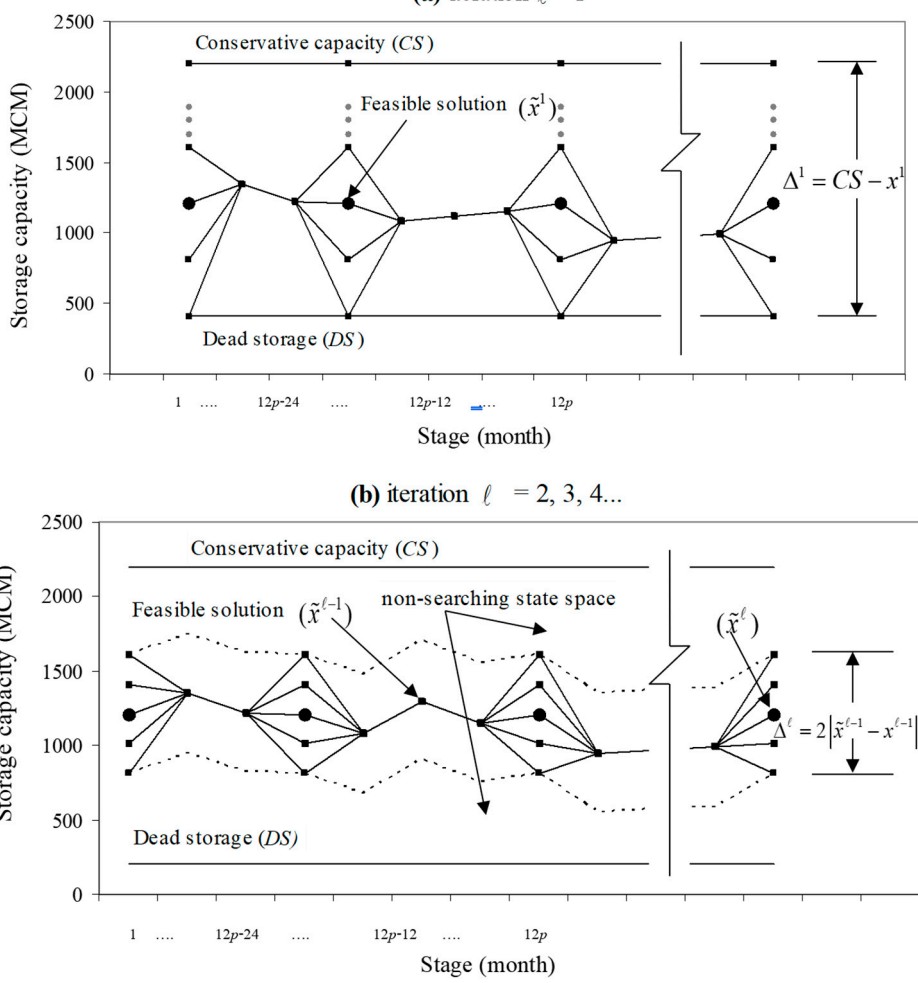

**Figure 8.** The computational dimensionality of DP/PPO to search for the optimal rule curves: (**a**) iteration l = 1 and (**b**) iteration l = 2, 3, 4, etc.

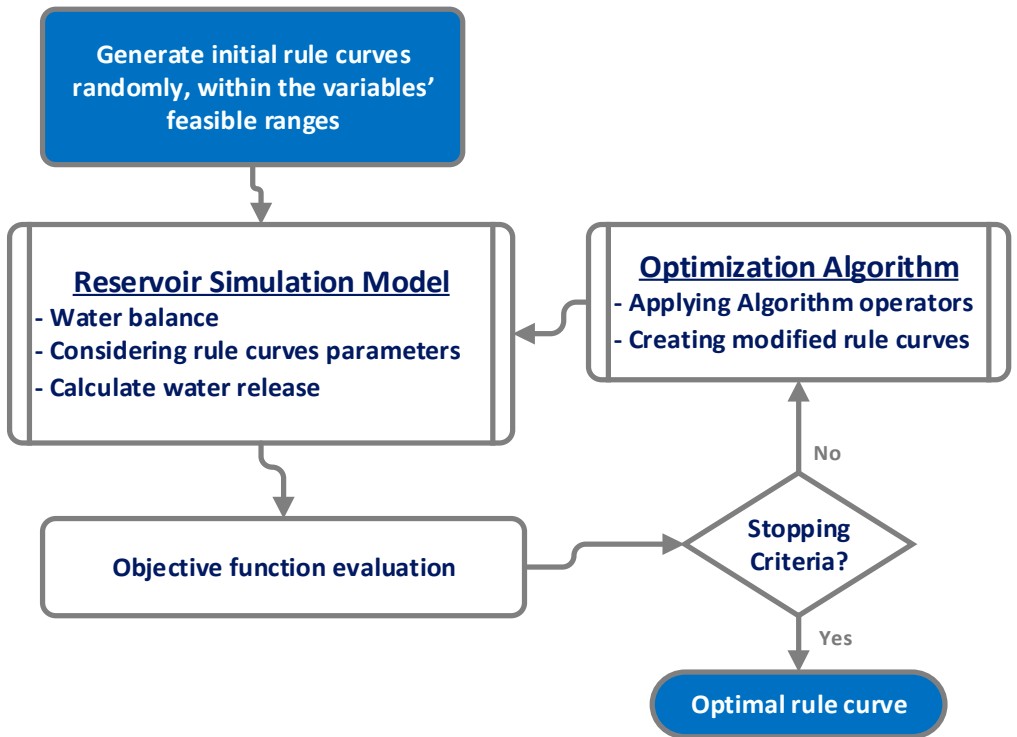

**Figure 9.** Heuristic and metaheuristic algorithms with reservoir simulation to search for the optimal rule curves.

### 4.3.2. The Shuffled Frog Leaping Algorithm

The shuffled frog leaping algorithm (SFLA) is a meta-heuristic optimization method inspired by the memetic evolution of a group of frogs when searching for food [54–56]. The SFL algorithm is derived from a virtual population of frogs in which individual frogs represent a set of possible solutions (possible rule curves). Each frog is distributed to a different subset of the entire population described as memeplexes. The different memeplexes are considered as different cultures of frogs that are located in different places in the solution space (i.e., global search). Each culture of frogs simultaneously performs an independent deep local search using a particle-swarm-optimization-like method [57]. To ensure global exploration [58], after a defined number of memeplex evolution steps (i.e., local search iterations), information is passed between memeplexes in a shuffling process. Shuffling improves the frog idea quality after being infected by frogs from different memeplexes, ensuring that the cultural evolution towards any particular interest is free from bias. In addition to improved information, random virtual frogs are generated and substituted in the population if the local search cannot find better solutions. After this, the local search and shuffling processes (global relocation) continue until defined convergence criteria are satisfied. Recently, a conditional shuffled frog leaping algorithm (CSFLA) integrated with a simulation model was applied to identify optimal reservoir rule curves [59–61].

The developed CSFLA for searching rule curves is described as follows. The CSFLA begins with an initial population of frogs, $F = \{X_1, X_2, \ldots, X_n\}$, created randomly within the feasible space. With the 24 decision variables for a single reservoir (upper and lower rule curves variables), the position of the $i$th frog is represented as $X_i = [x_{i1}, x_{i2}, \ldots, x_{i24}]^T$. Then, a set of rule curves is used in reservoir simulation and the release water is calculated by the simulation model considering these rule curves. Next, the release water is used to calculate the fitness function to evaluate the frog's position. The fitness function is the minimum of the average water shortage subject to constraints on the simulation model, as illustrated in Equation (4).

### 4.4. Evolutionary Algorithms

Some of the evolutionary algorithms that have been integrated with the reservoir simulation model to search for the optimal rule curves include the genetic algorithm (GA), differential evolution (DE), genetic programming (GP), and cultural algorithm (CA). Figure 10 shows the application of the evolution algorithm integrated with the reservoir simulation model to search for the optimal rule curves.

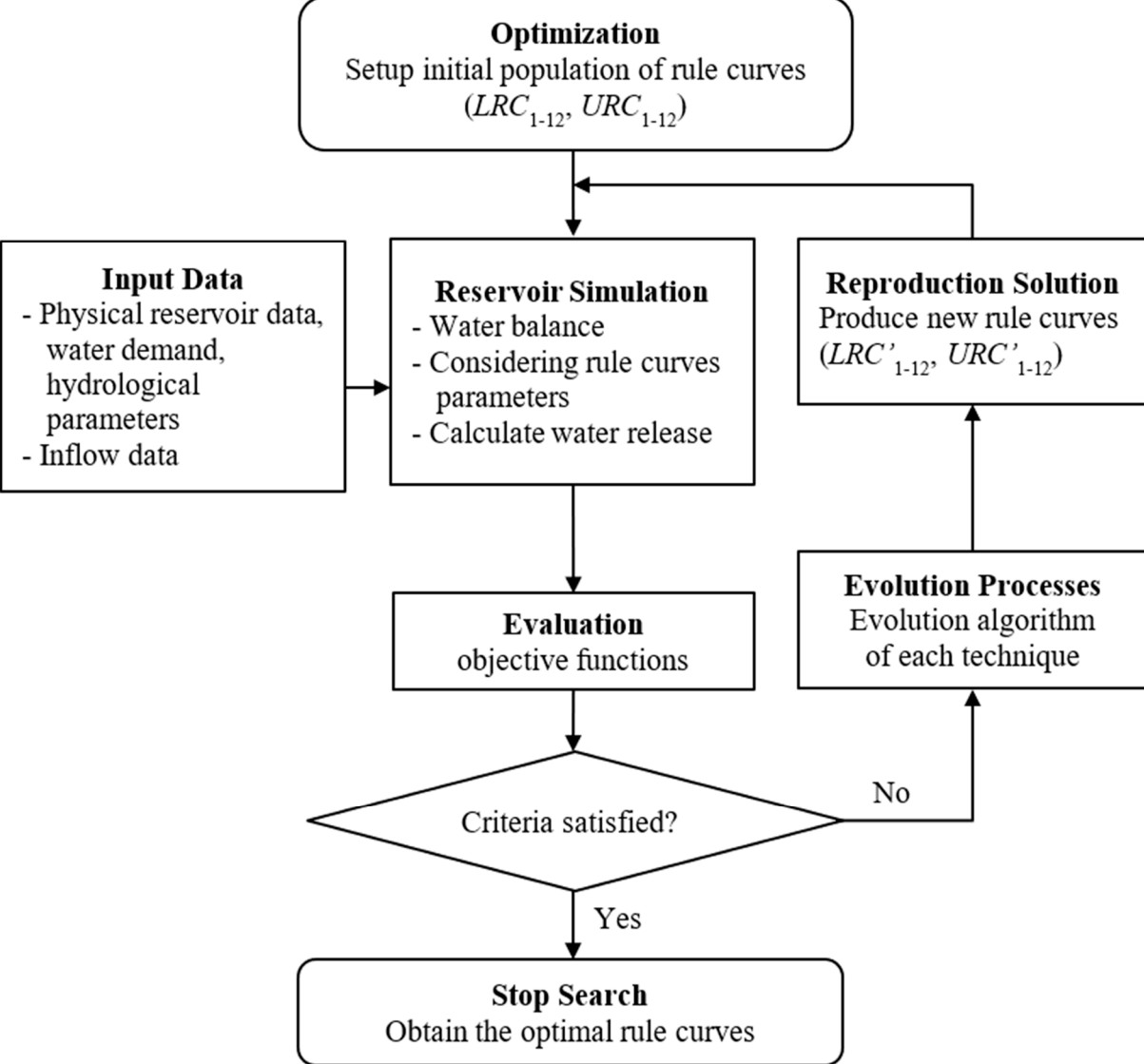

**Figure 10.** Evolutionary algorithm integrated with the reservoir simulation model to search for the optimal rule curves.

### 4.4.1. Genetic Algorithm

Genetic algorithms (GAs) have become increasingly popular over the past decade for solving various problems due to their robust performance [62]. In order to handle problem constraints and be integrated into simulation models, efficient computational methods have been presented in conjunction with GAs [63]. These methods have been successful in optimizing reservoir operations [64], which is a complex problem that involves the management of water release from a reservoir to meet various demands while taking into account multiple objectives, such as flood control, hydropower generation, and irrigation.

The GAs integrated with the reservoir simulation model involves transferring decision variables from the physical space of the problem to the computational space using a coding

approach. Each decision variable represents the value of the monthly rule curves, which is defined as the upper or lower bound. A population of coded solutions is randomly generated in the search space of the problem. The water release values from the reservoir at each time step are then calculated by the simulation model, and the performance criteria of the system in supplying reservoir targets are evaluated. In an iterative process, GAs operators, such as selection, crossover, and mutation, are executed on the existing population to produce new solutions. The simulation model simulates the rule curves and their performance using the generated solutions and estimates the released water, the performance criteria of the system, and finally the value of the predefined problem objective function. The process of reproducing new populations continues until the stopping condition of the algorithm is satisfied, and at the end, the best solution that is found represents the optimal rule curves of the reservoir.

GAs have been widely used in studies seeking optimal reservoir operation worldwide, including in Thailand [65–70]. For example, in the face of climate change and land use changes affecting runoff flows into reservoirs in upstream areas, the development of the optimal rule curves for large- and medium-sized multipurpose reservoirs has become necessary to improve their suitability and efficiency for future reservoir management [8,71,72]. In a recent study, the objective function was to minimize water shortages during the dry season and prevent overflows from reservoirs during the flood season. The results showed that the optimal rule curves generated by GAs can effectively reduce the frequency and volume of water shortage situations and water overflow, while maintaining a reasonable efficiency compared to the existing rule curves. The use of GAs in reservoir operation optimization can thus help decision-makers to make better informed decisions that ensure water security for all stakeholders.

### 4.4.2. Differential Evolution

Another efficient evolutionary search algorithm that has been successfully used in reservoir operation optimization and rule curve extraction studies is the differential evolutionary algorithm (DE) developed by Storn and Price [73]. In 2008, Ready and Kumar [74] applied a multi-objective functional model of DE to establish a water allocation policy for irrigation to support multi-crop systems under water scarcity situations that result in crop failure. A replication of this case study took place in Malaprabha Reservoir, India. Their results suggest that the proposed DE could be used to develop different strategies for irrigation planning and reservoir operation policies and to select the best possible solutions appropriate to the expected hydrological conditions. In Thailand, DE was connected to a reservoir model to develop the reservoir rule curves of the Lam Pao Reservoir, a large multipurpose reservoir located in Kalasin Province in northeastern Thailand. The results clearly indicated that the rule curves developed with DE can reduce the frequency, amount of water scarcity, and flooding during the dry and flood seasons better than the old rule curves. In addition, it demonstrated the development of appropriate reservoir operating rules to accommodate variations in hydrological conditions [75]. To date, DE has also been popularly applied to study optimum reservoir rule curves integrated with other algorithms. For example, hybrid DE and Bat algorithm (BA), enhanced differential evolution (EDE), and a differential evolution with particle swarm optimization (A-DEPSO) were used to solve the complex problems of a case study of four hydropower reservoirs with multi-reservoir operations [76–78].

### 4.4.3. Genetic Programing

Genetic programing (GP) describes a set of alternative techniques for application in a wide range of engineering problems [79]. Numerical methods have used GP for resizing structures in order to search for the optimal cross-sections and connecting the joints to achieve the minimum weight. GP has been applied to search for the optimal reservoir rule curves in the Huay Ling Jone reservoir, Yasothorn Province, Thailand [80]. Furthermore,

GP was used to develop a real-time reservoir operation considering the forecast of inflow flowing into the reservoir [81] and the reservoir's operating rule [82].

The process of GP begins with a randomly generated initial population of computer programs [83]. Each program in the population represents a parse tree generated by combining its functions (nodes) and terminals (leaves), which are suitable for the problem and defined in a function set and a terminal set, respectively [84]. The function set may include arithmetic and mathematical functions, conditional and Boolean operators, iterative functions, and user-defined functions or operators. The terminal set contains the arguments for the functions. Once the initial population is generated, the current population is repeatedly replaced with a new population (new generation) using genetic operators (reproduction, crossover, and mutation) until the population's best fitness reaches the desired value or the maximum number of generations is reached. The genetic operators used in GP are the same as those used in a basic genetic algorithm.

### 4.4.4. Cultural Algorithms

Cultural Algorithms (CAs) are a type of approach to evolutionary computational methods that was developed by Reynolds [85]. The cultural algorithm simulates social evolution based on learning agent-based modeling (ABM) techniques and based on experience and knowledge gained over time [86,87]. In addition to the population, this algorithm relies on another element called the belief space. The processes defined in the algorithm are very useful in improving the efficiency of the process to achieve the optimal solutions in the search space. In the approach, the population is the set of possible solutions in the search space, while the belief space is the space of knowledge obtained during the searching process. The accumulated data in the belief space are shared across the population and are applicable to all population-based optimization techniques [88].

Conditional Cultural Algorithm (CCA) was integrated with the reservoir simulation model in order to optimize the reservoir rule curves. In that study, the minimization of the average water shortage function was used as the optimization objective function. The CCA was applied with historical inflows, future inflows under scenario B2, water demand, and physical reservoir data to determine the optimal rule curves of the Huai Luang Reservoir in the northeast of Thailand [89].

In the first step, a reservoir simulation model is coupled with a conditional cultural algorithm. Computations start from randomly generating an initial population of reservoir rule curves, adjusting domain-specific knowledge, updating the belief space using acceptance function, and stopping criteria. The physical and initial data of the problem are considered, and then the reservoir rule curves in the initial population space are sent to the reservoir simulation model to evaluate their performances. Using the monthly streamflow, the objective function of the problem is calculated and the performance of the solutions in the population is determined [71]. After that, new reservoir rule curves are generated using algorithm operators and the initial population is replace if appropriate. This iterative process continues until the stopping condition is satisfied.

### 4.5. Swarm Algorithms

The swarm algorithms that have been applied with the reservoir simulation model for searching optimal rule curves include particle swarm optimization (PSO), cuckoo search algorithm (CS), firefly algorithm (FA), flower pollination algorithm (FPA), gray wolf optimizer (GWO), wind driven optimization (WDO), ant colony optimization (ACO), and honey-bee mating optimization (HBMO). Figure 11 shows the application of swarm algorithms integrated with the simulation model to search for the optimal rule curves.

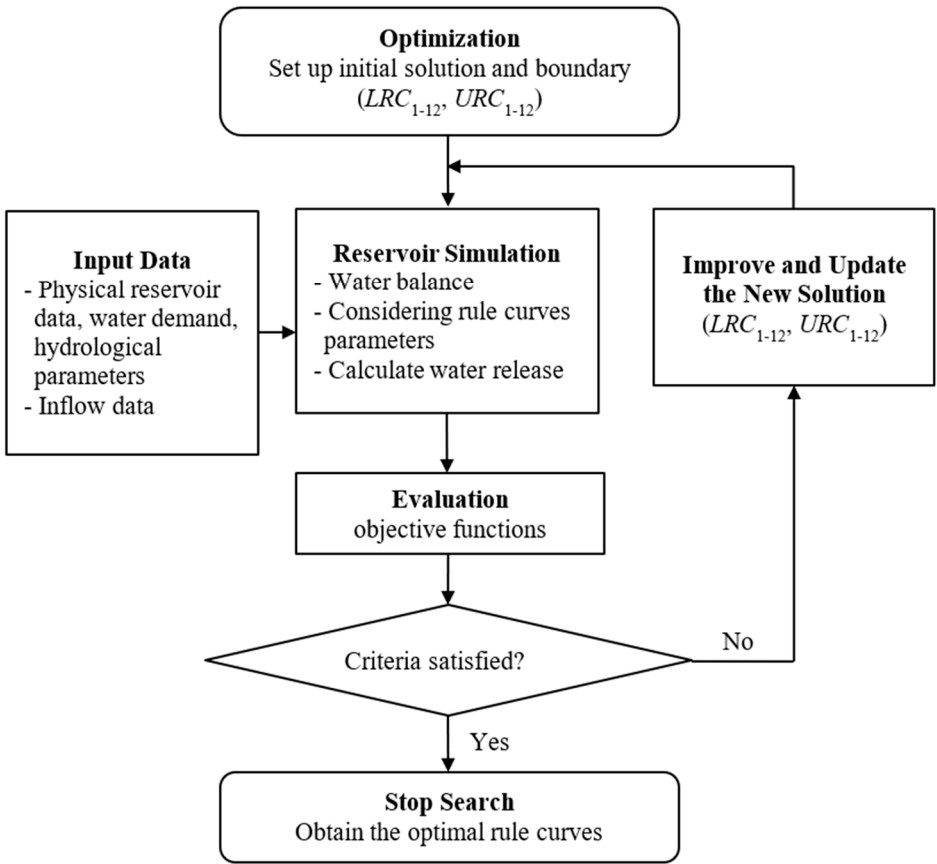

**Figure 11.** Swarm algorithms integrated with the reservoir simulation model to search for the optimal rule curves.

### 4.5.1. Particle Swam Optimization

The particle swarm optimization (PSO) algorithm, which was inspired by the swarm behavior of some organisms (such as birds and fish) and proposed by Eberhart and Kennedy [90], is one of the most efficient optimization methods to solve continuous problems. As with all guided stochastic search algorithms, the search begins with the generation of a swarm of random solutions in the search space, and then based on the rules defined in the algorithm, the swarm is updated in each iteration [91]. Although this algorithm is population-based, its operators have nothing in common with evolutionary methods and work based on swarm intelligence processes. Thus, in PSO, the potential solutions, or the particles, move towards the optimal point by following the optimal particles and using their personal experiences. PSO has been used in many fields, such as function optimization [92], artificial neural network and fuzzy system control [93,94], and reservoir operation planning and management [38].

Conditional particle swarm optimization (CPSO) with the reservoir simulation model has been applied to search optimal rule curves [95]. In Thailand, the Lampao Reservoir, which is located in the northeastern region, was used for the illustrative application [95]. In 2014, Zhang et al. [96] proposed an improved adaptive Particle agglomeration optimization (IAPSO) algorithm to address a number of conflicting objectives and limitations. This is related to non-linear problems with complex dynamic constraints, which are important in the context of reservoir system operation. The study was tested on the Three Gorges Project (TGP) and XiLuoDo Project (XLDP) power generation reservoirs in China, and IAPSO was compared with other algorithms, such as wavelet PSO (WPSO), adaptive PSO (APSO), and basic PSO (BPSO). It performed better with more efficiency and durability and appeared to be better in terms of power generation benefits and convergence efficiency [96]. In addition, PSO has been integrated into the reservoir system simulations that are co-operating to

satisfy from a set of different priority requirements by considering hedging rules, which are based on the level of monthly water retention. For the objective function of this study, it increased efficiency and reduced large deficits and assigned different weights to accommodate different demand categories [97]. In recent years, PSO has remained popular for its application to improve reservoir allocation criteria considering the conditions of iteration variables using water volumes for water supply and water spills [98] as well as maximum energy for short-term and long-term flood control [99].

### 4.5.2. Cuckoo Search

The cuckoo search (CS) optimization algorithm is one of the algorithms developed to solve non-linear and continuous optimization problems. This algorithm was introduced by Yang and Deb [100], inspired by the life of a family of birds called cuckoo. The cuckoo search algorithm is based on the optimal lifestyle and interesting features of this species, such as their egg-laying and reproduction characteristics. Adult cuckoos lay eggs in the nests of other birds. If the cuckoo eggs are not detected and destroyed by adult host birds, they will grow and become adult cuckoos. Under the influence of environmental characteristics, adult cuckoos migrate in groups hoping to find a suitable environment for life and reproduction. In this algorithm, the suitable environment is the global Optimum in the objective function of the optimization problem. To date, this algorithm has shown a good performance in various optimization scenarios. The female cuckoo inspects different nests to find a species of bird whose eggs, laid by them, are very similar in color and pattern to her own. Meanwhile, some species of cuckoos lay their eggs only in the nests of certain types of host birds. These cuckoos learn to lay eggs that closely replicate the color and pattern of the host's eggs. However, many host birds also learn to distinguish cuckoo eggs from their own. In such a case, either the cuckoo eggs are thrown out of the nest or the host bird abandons its nest and nests again in another place [101].

CS has often been applied in optimizations that seek to solve the problem of reservoir management in different conditions and variables. CS is considered an alternative to the optimal operation of multi-reservoir systems (OOMRS) for the purposes of maximizing energy production. Punitive methods are used to address physical and operational limitations [102,103]. The constraints of irrigation and flood control are considered [104], including the connecting conditional cuckoo search algorithms (CCS) and the reservoir simulation model, which has been applied to determine optimal reservoir rule curves [105]. This algorithm search begins with an initial feasible solution and boundary, as well as input data. The number of available host nests and the total number of iterations is set up. With 24 decision variables per reservoir (rule curve variables for the upper and lower rule curves), a nest for the cuckoos is represented as $Xi = [xi_1, xi_2, \ldots, xi_{24}]$. Then, a set of rule curves is used in the reservoir simulation and the release water is calculated by the simulation model using these rule curves. The release water is then used to calculate the fitness function to evaluate a nest. A nest is then chosen randomly. A new set of rule curves is used in the reservoir simulation and the release water is used to calculate the fitness function again for $Z(Xi+1)$. Next, the fitness function $Z(Xi+1)$ and the fitness function $Z(Xi)$ are compared: if $Z(Xi+1)$ is larger than $Z(Xi)$, return and choose a new nest, but if $Z(Xi+1)$ is smaller than $Z(Xi)$, replace $Xi$ by $Xi+1$ and keep the new nest (accepted rule curves) for the next iteration. The next iteration is performed by choosing the new nest if the termination criterion is not satisfied. The process is then continued until the criterion is satisfied, as illustrated in Figure 11. In the case of climate change impact studies, CS has been used to optimize the multifunctional performance of reservoir systems. Its purpose is to meet downstream water needs and control potential flooding [106].

### 4.5.3. Tabu Search Algorithm

The tabu search algorithm (TSA) is a meta-heuristic procedure designed to search for an optimal solution. This algorithm is different from other meta-heuristics that do not rely on randomness or selection based on probability. It is a deterministic method that searches

for solutions from the immediate best solution. It has been adapted to solve many problems in engineering, such as civil, industrial, electrical as well as water resources [107–109]. The tabu search is recognized as being able to avoid provide the final solution that is the local optimum value and can continue to search until the solution is near to the global optimum value [110,111].

From 2018 to 2020, the conditional tabu search algorithm (CTSA) integrated with the reservoir simulation model was applied to find the optimal rule curves. The minimum average water shortage per year was used as the objective function for the finding procedure, including hydrologic data, water demand, physical reservoir data, and future runoff using 50-year future climate data. The proposed model was applied to determine the optimal future rule curves of the Ubolratana Reservoir in the northeast region of Thailand [8,112,113]. In addition, TSA has also been studied for the solution and development of optimal reservoir operating policies and has been integrated with other algorithms, for example, in flood protection [114], the interoperability scheduling solutions of large reservoirs that require a navigation ship lock waiting time under multiple constraints, and the long-term energy scheduling of hydropower systems [115,116].

### 4.5.4. Firefly Algorithm

The firefly algorithm (FA) is a swarm-based metaheuristic algorithm inspired by the flashing behavior of fireflies. FA is one of the algorithms released in 2009 by Yang [117]. The algorithm imitates the behavior of fireflies and has ideal rules. One of these rules is that an individual firefly will follow a brighter firefly and, if there is no brighter firefly, the firefly will move randomly [117]. It is an effective and easy-to-implement algorithm [118]. FA has been popularly applied in studies to solve reservoir management problems in both single and multi-purpose reservoir systems, with the main objectives being irrigation water supply [119] and irrigation and hydropower generation [120]. It is also effective in finding suitable solutions to a continuous reservoir problem [121]. Moreover, FA was applied with the reservoir simulation model for searching optimal reservoir rule curves in a flood control area. This algorithm search begins with an initial feasible solution and boundary and input data. The 24 decision variables per reservoir are represented by both the upper and lower rule curves. Then, a set of initial rule curves is used in the reservoir simulation and the release water is calculated. The release water is then used to calculate the fitness function to evaluate the initial rule curves. Then, a new set of rule curves is created and used in the reservoir simulation. The release water is used to calculate the fitness function again and the new set of rule curves is evaluated. Next, the fitness function for this iteration and previous iteration are compared to accept the best one. The next iteration is performed by again creating the new set of rule curves if the termination criterion is not satisfied. The process is then continued until the criterion is satisfied, as illustrated in Figure 11. The proposed model was applied to determine the optimal flood rule curves of the Nam Oon Reservoir in the northeast region of Thailand. The minimum average excess water per year and minimum frequency excess water were used as the objective functions [122].

### 4.5.5. Flower Pollination Algorithm

One of the alternative techniques that was recently adapted to find appropriate values is the flower pollination algorithm (FPA), which is based on the pollen grains of the flower. Each flower has a different way to lure a bird or insect pollinator to ensure pollination for reproduction and survival [123–125]. FPA is a highly effective technique that is suitable to search for the optimal reservoir rule curves. There were two objective functions that are considered to the search process: the minimum average water deficit and minimum average excess water. The application of FPA to develop reservoir rule curves has been studied in medium and small reservoirs in Thailand. The Huay Sabag, Huay Ling Jone, and Num Oon reservoirs, which are located in the northeast region of Thailand, were considered in one case study [126]. The operating process consists of the following parts. The connection of the FPA and reservoir simulation model starts with input data and all

the initial necessary data, such as dead storage level, normal high-water level, full capacity level, and monthly water requirements. The FPA procedure is based on the flowers and the pollination process. For this study, each decision variable was represented by the monthly rule curves, which are defined as the upper rule and lower rule curves. After the first set of flowers in the initial population were calculated, the monthly release water was calculated by the simulation model considering those rule curves. Next, the released water was used to determine the objective functions. Then, the evaluation of the criteria was undertaken to evaluate the objective function. After that, the reproduction process creates new rule curve values in the next iteration to find the current best solution. This procedure is repeated until all the values of the rule curves are appropriate.

Recently, several researchers have used FPA to assess the problems of various reservoir management schemes, both for single and multi-reservoir systems, as well as for specific purposes, for example, optimizing hourly scheduling for crop water allocation and hydropower generation [127] and the operation of a large hydropower development [128].

4.5.6. Gray Wolf Optimizer

Another recently developed method of finding suitable interest values is the new metaheuristic method that has been used to determine appropriate water allocation criteria, called the gray wolf optimizer (GWO) [129–131]. In this method, inspiration comes from the gray wolf and the search methods to improve the surrounding position and hunting of gray wolves. It is based on the hunting behavior of gray wolves, which require skill and ability to search for prey and to surround it [132,133]. From 2019 to 2022, several studies reported the application of GWO to search for the optimal solutions to reservoir operation problems. In 2019, Choopan and Emami [134] presented the application of GWO to predict water storage in the Shaharchay Reservoir in northwestern Iran. Its objective is to manage risks from flood and drought events. The results from the GWO technique showed statistically good predictions compared to evolutionary algorithms, such as the continuous genetic algorithm (CGA) [134]. Later, in 2020, at the Golestan Dam in Golestan Province, Iran, Donyaii et al. [135] developed a multi-objective gray wolf optimization (MOGWO) algorithm to find optimal reservoir operating criteria under changing climatic conditions. The associated objective functions defined in the optimization process are the risk reduction and maximization of model reliability indices under baseline conditions and climate change periods. In 2021, Niu et al. [136] presented a hybrid gray wolf optimizer (HGWO) to improve the optimum operation of real-world hydropower systems with the goal of maximizing the benefits from total electricity generation. The simulations indicated that the HGWO method produced more satisfactory scheduling plans than the multiple control methods.

Most recently, in 2022, Masoumi et al. [137] released the shuffled gray wolf optimizer (SGWO), a hybrid optimization algorithm inspired by the shuffled complex evolution (SCE-UA) algorithm and GWO to solve mathematical benchmark function and multiple reservoir performance optimization problems with different scales. Two hypothetical reservoir systems 4 and 10 and the Dez Dam in Iran, considered as a single reservoir system, were chosen as case studies in this research. In addition, the performance of the SGWO algorithm was compared to that of well-known evolutionary algorithms, such as particle cluster optimization (PSO) and genetic algorithm (GA). Their results showed that SGWO was able to achieve significantly better outcomes by using a significantly smaller number of function assessments. In Thailand, GWO with a reservoir simulation model to search for the optimal rule curves was applied. This algorithm search begins with an initial feasible solution and boundary and input data. The decision variables are represented the upper and lower rule curves. Then, a set of initial rule curves is used in the reservoir simulation. The release water from reservoir simulation model is then used to calculate the fitness function for evaluating the initial rule curves. Then, a new set of rule curves is created and used in the reservoir simulation. The release water from the reservoir simulation model is used to calculate the fitness function again and evaluated the new set of rule

curves. Next, the fitness function of this iteration and previous iteration are compared to accept the best one. The next iteration is performed by creating the new set of rule curves again. The process is then continued until the criterion is satisfied, as illustrated in Figure 11. The Ubolratana reservoir in Khon Kaen Province, the Lampao reservoir in Kalasin Province, and the Nam Oon reservoir in Sakon Nakhon Province were considered as the case studies [138].

### 4.5.7. Wind-Driven Optimization

The wind-driven optimization (WDO) algorithm is an alternative optimization technique that has recently gained popularity. The algorithm is a nature-inspired global optimization methodology that draws inspiration from atmospheric motion. Essentially, WDO is an evolutionary adaptation of air parcels in the atmosphere that are able to find the best pressure to balance the atmosphere [139]. WDO is a population-based iterative heuristic global optimization algorithm where a population of infinitesimally small air parcels navigates over an N-dimensional search space following Newton's second law of motion, which is also used to describe the motion of air parcels within the Earth's atmosphere [140]. The technique has been successfully applied to optimize electromagnetics problems [141–144], and is particularly suited for solving problems with both discrete and continuous parameters.

WDO has also been applied to search for optimal reservoir rule curves in flood control areas. For example, the technique was used to determine the optimal flood rule curves of the Nam Oon Reservoir in the northeast region of Thailand, with the minimum average excess water per year and minimum frequency excess water used as the objective functions [145]. WDO has also been used to develop rule curves for multiple reservoir systems, with the objective of improving the existing method of managing a single reservoir that caused serious problems during the rainy season when it was flooded in 2017 in Sakon Nakhon Province, Thailand [146]. In both studies, the WDO technique was combined with a reservoir simulation model. The process involves starting with an initial population that is created randomly within the feasible space, which is the value between the dead storage capacity and the normal capacity levels of the reservoir. Each decision variable represents the monthly rule curves in the reservoirs, which are defined as the upper level and lower level.

### 4.5.8. Ant Colony Optimization

The ant colony optimization (ACO) algorithm was developed as another search optimization technique that was motivated by the natural phenomenon of ants depositing pheromones on the ground to mark favorable paths that should be followed by other members of the colony [147]. ACO is a probabilistic technique to solve computational problems that can be reduced to finding a good path through graphs and has been widely applied to various problems [148,149]. ACO emerged as useful for improving the operating rule of the Hirakud multipurpose reservoir system in India. During the implementation, a limited time series of inflows was considered as well as the classification of reservoir volumes into several classes and the consideration of reservoir discharges for each time period according to the pre-determined criteria of suitability. In this case study, multiple objectives were identified, including reducing flood risk, minimizing irrigation deficits, and maximizing hydropower generation as a priority. The final objective was to implement the developed model into the monthly reservoir operation [150]. ACO has been implemented to find a solution to the problem of hydropower reservoir operation that is defined as a partially connected graph consisting of a set of rule curves connecting the nodes of the graph. Each rule curve was assigned to represent the local operating policy of the reservoir. The results clearly indicated that ACO paves the way for the efficient utilization of the incremental solution generation mechanism available in ACO to enforce explicit problem constraints [151].

Then, in 2013, ACO was developed by identifying constraints so as to address the appropriate multi-reservoir operating system problem by considering the decision parameters of the problem as well as its responsiveness to both the storage and discharge volumes. The algorithms developed are known as constrained ant colony optimization algorithms (CACOAs). The results were found to outperform the conventional unrestricted ACOs, especially in the ability of CACOAs to quickly and optimally solve multiple reservoir operational problems [152]. Moreover, Kangrang and Lokham [153] also proposed a conditional ACO (CACO) linkage to a reservoir model to find a suitable reservoir control curve by selecting the case study of the Lam Pao Reservoir in Kalasin Province, northeastern Thailand. In their study, the objective function of the search was identified as the lowest average water scarcity and the increase in water allocation for future irrigation areas was identified. Finally, the comparison of the new control curve developed with CACO to the existing one suggests that it is effective in reducing the water shortages that are appropriate to seasonal hydrological conditions, especially regarding the reservoir inflow. Overall, the strength of the ACO is its quick search for global solutions as well as the appropriate application to various conditions. Accordingly, ACO is popularly used to find solutions in water resource management [154].

### 4.5.9. Honey-Bee Mating Optimization

The honey-bee mating optimization (HBMO) algorithm is a nature-inspired algorithm that mimics the mating process of honey bees. In recent years, HBMO has shown promise in solving reservoir management problems by being applied to highly constrained and/or unconstrained real-value mathematical models with the objective of minimizing the total squared deviation from the target requirement. Comparisons with other known heuristic methods have been favorable [155,156]. Researchers have used HBMO to extract the monthly linear operating rules of reservoirs for irrigation and hydropower purposes, considering decision variables such as reducing the water supply deficit [157]. In 2011, an improved version of HBMO was introduced to develop operating rules for multi-reservoir systems with the objective of improving the release rule and the retention-volume-balancing function, which form the operating policy. The proposed rule curves were able to manage the tight constraints that define parallel reservoir operation in such a way that all generated solutions are possible after a particular set of iterations [158]. While still a relatively new algorithm, HBMO has shown promise in the field of reservoir management and could potentially offer significant benefits in optimizing reservoir operations.

In addition, Solgi et al. [159] introduced a new enhanced HBMO (EHBMO) that relies on a new mating process, replacing the one used in the HBMO algorithm. This change enables EHBMO to achieve a solution that is as close as possible to the global optimum with less computational effort compared to HBMO. The performance of the EHBMO algorithm was tested with constrained and unconstrained mathematical optimization problems. It was also used to find the optimal functioning of multi-reservoir systems. Recently, in Thailand, HBMO has been applied to create the optimal reservoir rule curves impacted by future climate change between 2020 and 2049 based on climate data from CMIP5 with the scenarios from RCP4.5 and RCP8.5, respectively. The case study examined the effectiveness of new rule curves in managing fluctuations in streamflow at the Ubolratana Reservoir, located in the northeast region of Thailand. The results demonstrated that the new rule curves were successful in addressing variations in streamflow and showed promise for future water management efforts in the region. Specifically, the new rule curves were found to increase the storage capacity of the reservoir during the flood season and reduce water shortages during the dry season. These findings were compared to the existing rule curves, and the results showed clear improvements with the implementation of the new curves [160]. By demonstrating the positive impact of these new rules in a real-world setting, this study has important implications for future water management practices in Thailand and beyond. Hence, the methodology and results of this study are presented in this paper in Section 5.1.

## 5. Suitable Future Rule Curves

Finally, in addition to the various optimization techniques presented previously, the development of an optimal reservoir rule curves by applying correlated hydrological data is also presented as well as its implications for reservoir rule curves operation management. In particular, the development of a reservoir rule curves requires the volume of water flowing into the reservoir originating from the upstream area above the reservoir, which is regarded as the main resource of the stored water in the reservoir [161]. However, because of the global climate change problem, the amount of rain and temperature fluctuates considerably when compared to past values and is likely to become more severe in the future. For this reason, global climate change is expected to have an inevitable, direct impact on the hydrological system. Moreover, the problem of land use changes due to the encroachment of watershed forest areas due to the demand for agricultural areas and human habitation has increased and tends to increase according to the growth of the economy. These situations have affected runoff flow conditions, in particular, surface water behavior, evaporation, interception, and infiltration [162,163]. In addition, the implementation of a stakeholder participation process in reservoir management [164], consisting of highly skilled and experienced professionals in controlling water allocation appropriately in each season and in flood or drought situations, has been taken into consideration to improve the reservoir rule curves after its development from different models.

The results of work in this section are, therefore, expected to make the reservoir rule curves developed through this multi-stakeholder participatory process more appropriate when integrated with the approaches described in first two sections. Therefore, the problem of climate and land use changes affects reservoir management, from the perspective of the operation starting from the analysis of hydrological conditions to the last dimension, which is the reservoir management model that has been reviewed by highly skilled and experienced experts [165]. For this reason, in this section, we present the development process of the reservoir rule curves taking into account global climate change (especially precipitation and temperature) and land use changes. It is based on reliable numerical models that are widely used to simulate future situations [71,166]. It also includes a final improvement with the participation of experts, leading to the development of an efficient reservoir control curve suitable for future hydrological variability scenarios.

### 5.1. Climate Change

The study of climate change and its potential impact on hydrological systems is a complex and important field of research. To estimate future climate change, researchers often use climate models, such as global circulation models (GCMs) and regional climate models (RCMs) [167]. In Thailand, for example, the study of reservoir rule curve development has used future climate data integrated with hydrological models to estimate future runoff flowing into reservoirs.

One RCM that has been widely used in Southeast Asia, including Thailand, is the Providing REgional Climates for Impacts Studies (PRECIS) model, developed by the Hadley Center for Climate Prediction and Research. PRECIS uses data from the global climate model dataset ECHAM4 as a base for its calculations [168]. The model has a spatial resolution of 0.22 °C in the grid, which is downscaled to 0.2 °C or about 20 km [169,170].

The goal of these studies is to apply the results of future climate predictions from models such as PRECIS to assess runoff into reservoirs and develop suitable control curves for managing water resources in the future. For example, a recent study by Kangrang et al. [9] used the results of climate change forecasts for the A2 scenario from 2015–2064 to assess the impact on the Ubolratana Reservoir in the northeast region of Thailand. The study found that, while rainfall trends for the Loei station were lower than the baseline year data during the first 30 years of the A2 scenario, after 40 years, the average rainfall is expected to be higher than the baseline year. In contrast, both the Ubolratana Dam and Chulabhorn Dam stations are expected to experience an increase in rainfall trends.

However, the study also found that the maximum and minimum temperatures are expected to increase for all three climate stations compared to the baseline year. This information can be considered as an analysis of changes in hydrologic conditions in the upstream area of the Ubolratana Dam (see Figure 12). Moreover, changes in land use are expected to impact the variability of runoff flowing into the reservoir, highlighting the need for the ongoing monitoring and management of water resources in the face of climate change.

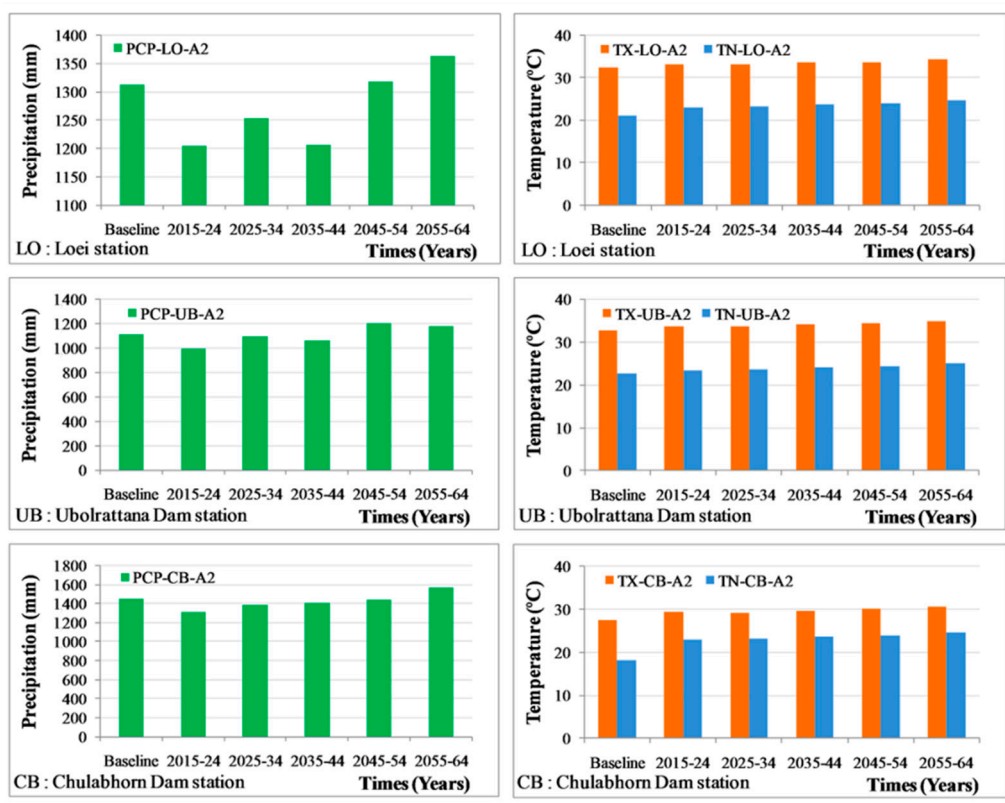

**Figure 12.** Future climate trends compared to the baseline A2 scenario [9].

### 5.2. Land Use Changes

Land use and land cover changes can have a significant impact on runoff in a given watershed area. Because of this, there are numerous methods used to monitor these changes, ranging from less precise methods to more accurate ones. In general, land use and land cover changes are represented in maps over a certain time period, such as 1 year, 2 years, 5 years, 10 years, or 15 years. The length of the time period is usually determined based on the objective of the analysis and the functional unit being used.

One model used for decision-making that incorporates aspects of Cellular Automata (CA) and Markov Chain is called CA Markov [171,172]. This model is used to predict land use and land cover changes and has been applied to a variety of water resource problems, such as runoff analysis due to rapid climate change and urbanization [173], flood evaluation [174], and soil erosion. Furthermore, there has been extensive utilization of the CA Markov model in anticipating future land use and land cover transformations [175,176].

In this section, we explore the results of a study that analyzed land use scenarios in the upstream area of the Ubolratana Reservoir, which is located at the outlet point in the east of the watershed and is also the climate change forecast area. In order to accurately assess the variation in runoff, we need to study the two variables (climate and land use patterns) in parallel and in a dynamic manner [177,178].

To forecast the patterns of future land use, the land uses in the past were used as the baseline. The land use types were classified based on the criteria of functional unit. An example of 12 land use types is shown in Figure 13. The future land use map created from

this analysis can be used to estimate the future inflow into the reservoir, providing valuable information for water resource management and planning.

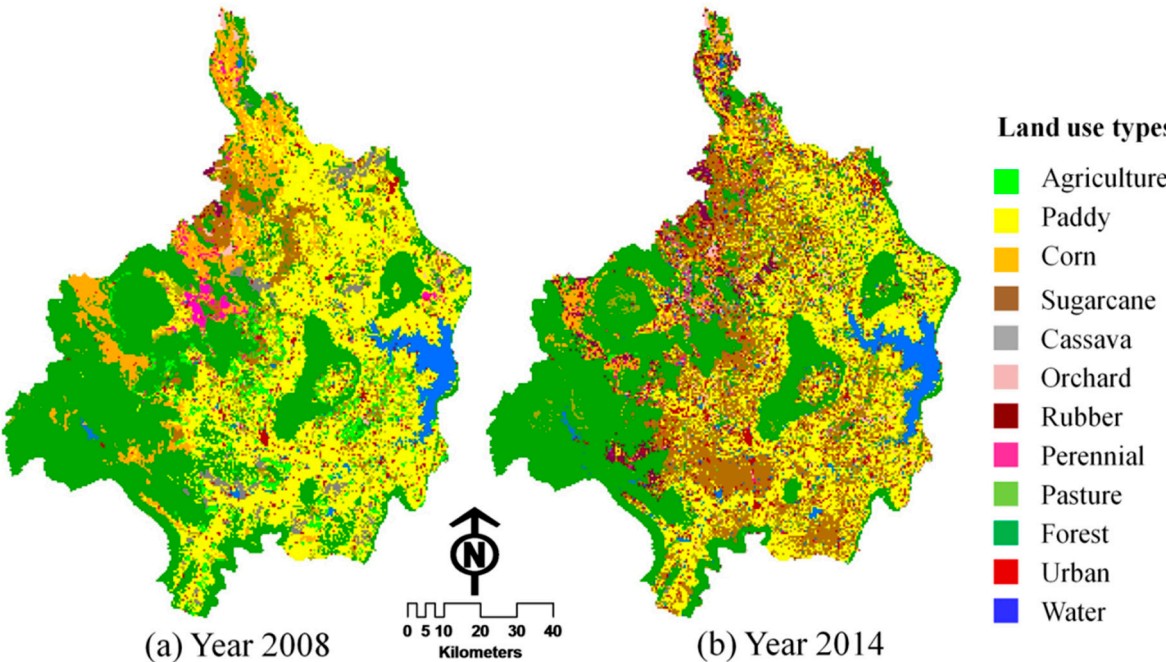

**Figure 13.** Baseline land use maps: (**a**) 2008 and (**b**) 2014 [9].

In this section, the study results of simulated future land use maps using the CA Markov model are discussed. The simulation was conducted for the period of 2015–2064 and compared to the baseline period between 2008 and 2014. The analysis revealed that the largest land use changes were the expansion of sugarcane and rubber tree, while paddy field and forest areas decreased. To illustrate the transition areas, five simulated land use maps are presented in Figures 14 and 15. These maps provide a visual representation of the changes in land use patterns over time and can be useful in predicting the future inflow into the reservoir.

The projected results from these two sections was integrated with SWAT hydrological models to assess variations in runoff volume into the reservoir in future time series. Finally, runoff data obtained from the SWAT model was used to construct reservoir rule curves that are appropriate for the variability of hydrological conditions.

*5.3. SWAT Model*

In the past decade, a simulation of the hydrological situation has been accepted that assess hydrological conditions using mathematical models in past, present, and future situations. The results are expected to be affected by climate change and there will be correlated changes in land use. One model widely used by hydrologists, researchers, and environmental and water resources engineers is the semi-distributed model called SWAT (Soil and Water Assessment Tool) [179]. SWAT is a hydrological model that was designed to assess the impact on water resources of rainfall and land use changes and operates on a daily time step. In addition, it can also work effectively with minimal input data, which is particularly suitable for areas that have limited default data. SWAT has been used widely to analyze runoff into a small basin [180–182]. The results of SWAT make it possible to recognize future trends in the runoff into the reservoir, which has the effect of enabling the prediction of the reservoir operation [183–185].

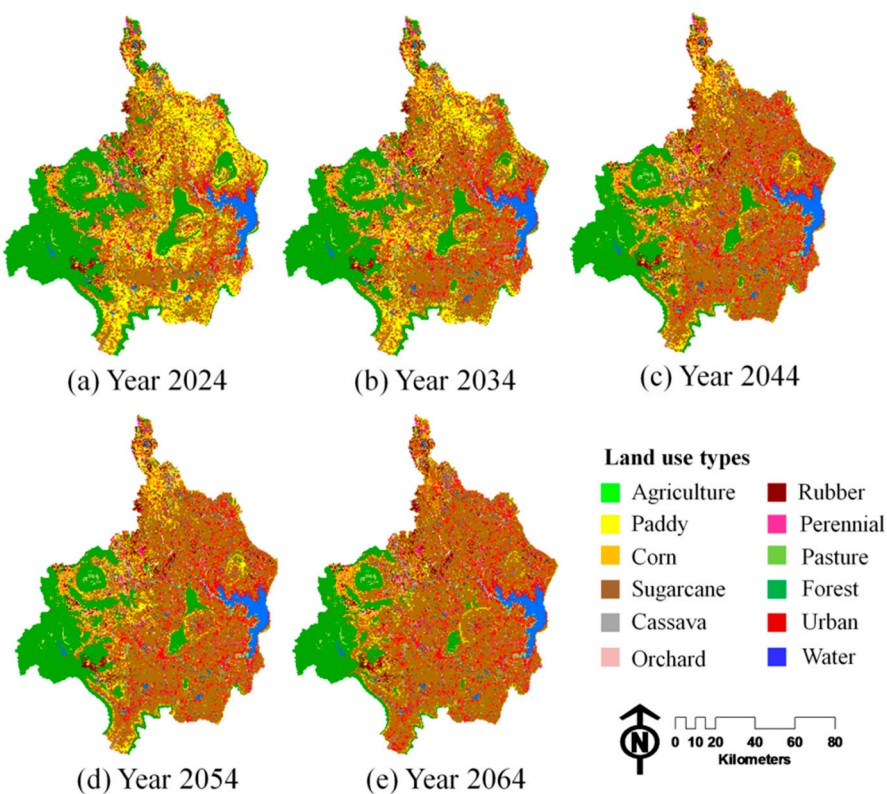

**Figure 14.** Simulated land use maps for 2015–2064: (**a**) 2024, (**b**) 2034, (**c**) 2044, (**d**) 2054, and (**e**) 2064 [9].

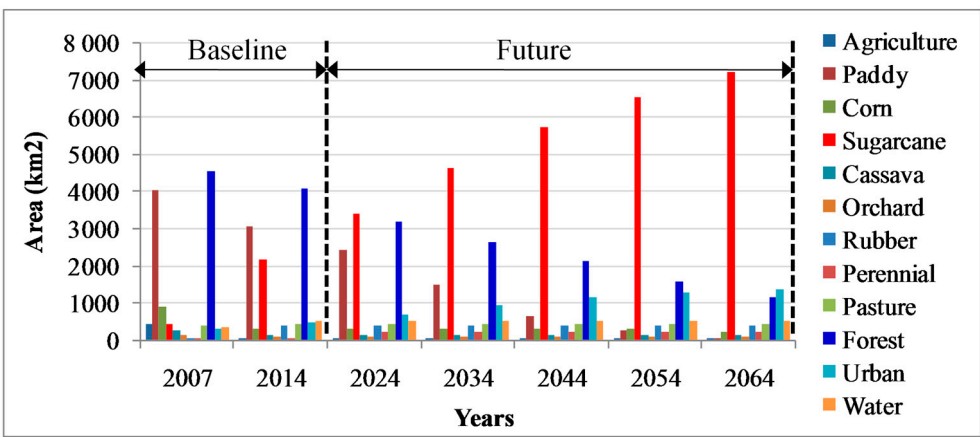

**Figure 15.** Baseline and simulated land use areas.

For the case study of the Ubolratana Reservoir presented in this section, in the initial implementation phase, SWAT requires input basic data, including DEM, stream lines, daily climate data, and spatial maps of land use and soil types. Then, the model defines the watershed delineation and calculates the simulation runoff during the baseline year. The effectiveness of the estimation is then considered by comparing the runoff from the observed stations and the runoff from SWAT as present in terms of $R^2$ (coefficient of determination), $RE$ (relative error), and $E_{ns}$ (Nash–Suttclife simulation efficiency). In this study, the calibration of SWAT was conducted by adjusting the eight hydrologic parameters: Alpha_BF, Gwqmn, Gw_Revap, Sol_Awc, Epco, Esco, Ch_N2, and Gw_delay.

The suitably calibrated SWAT with the hydrologic parameters, as compared with the record data, can then be used to forecast future runoff. Then, the simulated daily data from the PRECIS model and future land use from CA Markov were used in SWAT to estimate

the future runoff. The considered durations for the calculations were separated into five time periods of 10 years. The procedures in the calculation are shown in Figure 16.

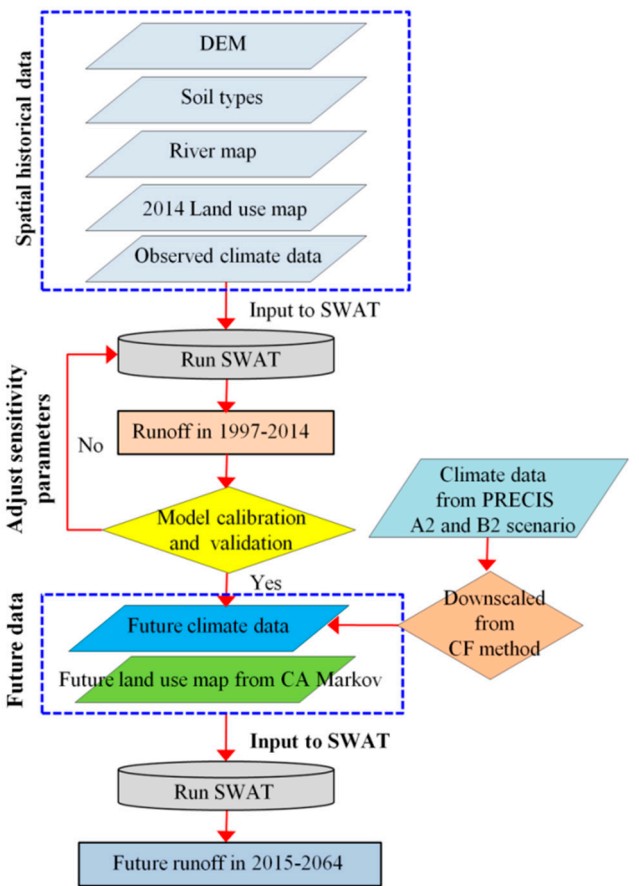

**Figure 16.** Study schematic of the calculation steps for future runoff forecasting.

The runoff of the SWAT model and runoff of the observed stations showed that the effective indexes ($R^2$, $RE$, and $E_{ns}$) could be accepted. In order to predict the future runoff at the Ubolratana Dam station between 2015 and 2064, the calibrated SWAT was simulated using inputs from the climate data of PRECIS, which also had decreased tolerances, and the land use maps of the CA Markov model. The simulated outcomes showed that the future average annual runoffs from the A2 and B2 scenarios were 4028.91 and 4580.50 MCM, respectively. Figures 17 and 18 indicate an increase in the runoff into the Ubolratana Reservoir in the future periods relative to the baseline period. Subsequently, SWAT-derived runoff was used as input for the development of reservoir rule curves linked to the reservoir simulation models and various optimization techniques were used to solve problems by searching optimal solutions based on many constraints and objective functions.

*5.4. Participation of Stakeholders*

The process of developing reservoir rule curves involves obtaining suitable curves from simulations with various algorithms and then improving them through stakeholder participation. In Thailand, the Royal Irrigation Department (RID) [186] controls the management of reservoirs, and the rule curves developed from the model require some improvement through a participatory process. To enhance the reservoir operating simulation for reducing floods in the Shell Mouth Reservoir on the Assiniboine River in Canada, Ahmad and Simonovic [187] used expert involvement approaches with the system dynamics (SD) method. They found that the operator trusted the model's application and was willing to help to enhance the SD model's structure to find a workable solution to the problem.

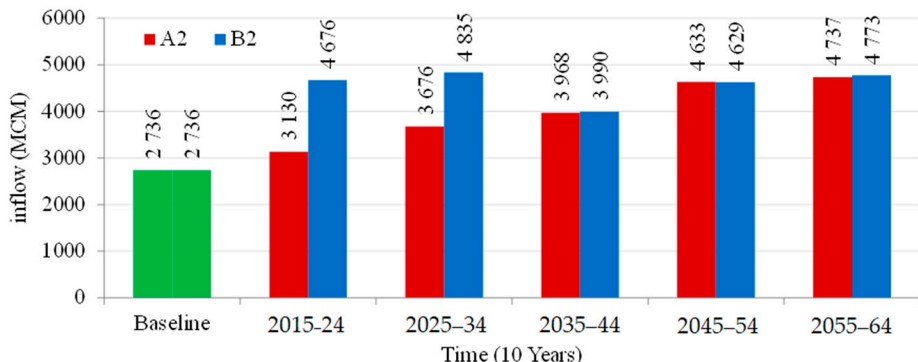

**Figure 17.** Baseline and future yearly Streamflow into the Ubolratana Reservoir [9].

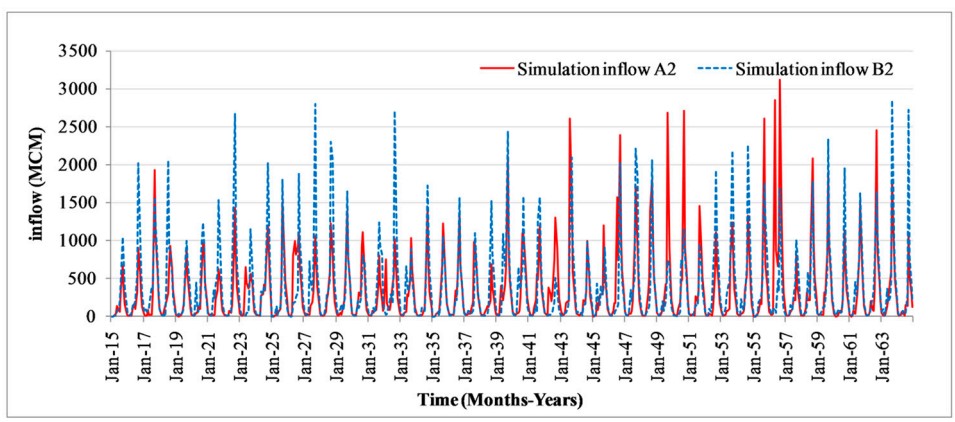

**Figure 18.** Future runoff into the Ubolratana Reservoir [9].

Similarly, Kangrang et al. [188] used expert participation to improve reservoir rule curves in Thailand using the conditional differential evolution (CDE) optimization approach. The findings showed that the appropriate rule curves of the Expert-CDE approach had efficiencies that may decrease water scarcity and overflow. Skilled operators, such as the director of reservoir operations, senior operations engineer, and technical operations engineer, were involved in the process, which consisted of surveying, observation, interviewing, a focus group, and a workshop [188]. Their ideas were implemented to rerun the simulation model and evaluate the objective function, and their recommendations were used to modify the final rule curves.

The results showed that the patterns of the accepted rule curves obtained using expert adjustment were smoother than the ones without adjustment, which was due to the rule curves being adjusted by the operators based on their experiences. The adjustments from the participation process led to the stakeholders accepting the rule curves as they understood the process, making it easier for operators to use them (see Figures 19 and 20). Although the results of using the rule curves with expert operators were close to the results of using the rule curves without adjustment for both scenarios, the rule curves with expert operators were still more accepted to be used in actual reservoir operation. This highlights the importance of stakeholder participation in developing reservoir rule curves that can be practically applied by the agency in charge [189].

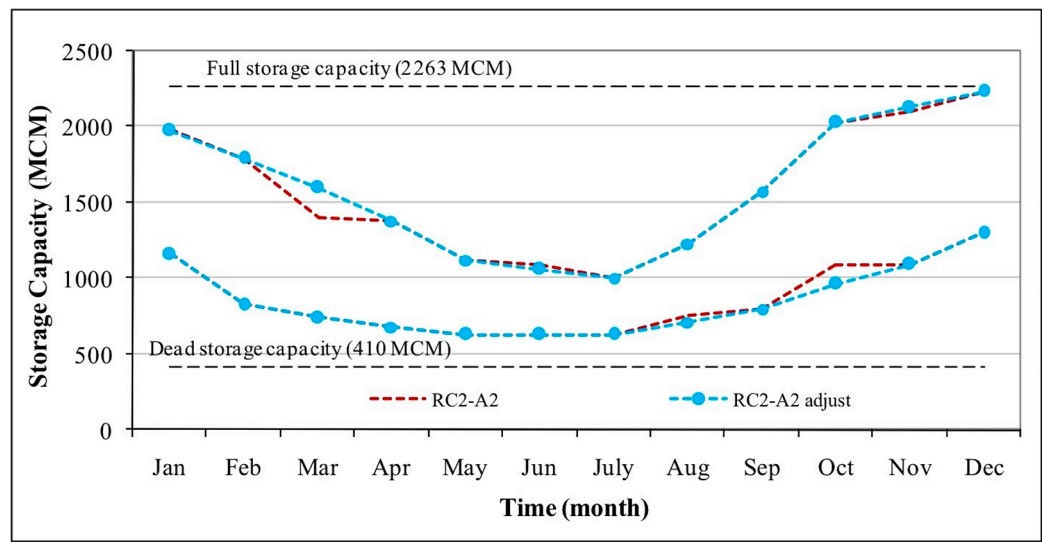

**Figure 19.** Optimal rule curves of the Ubolratana Reservoir (A2 Scenario) adjusted after stakeholder participation [9].

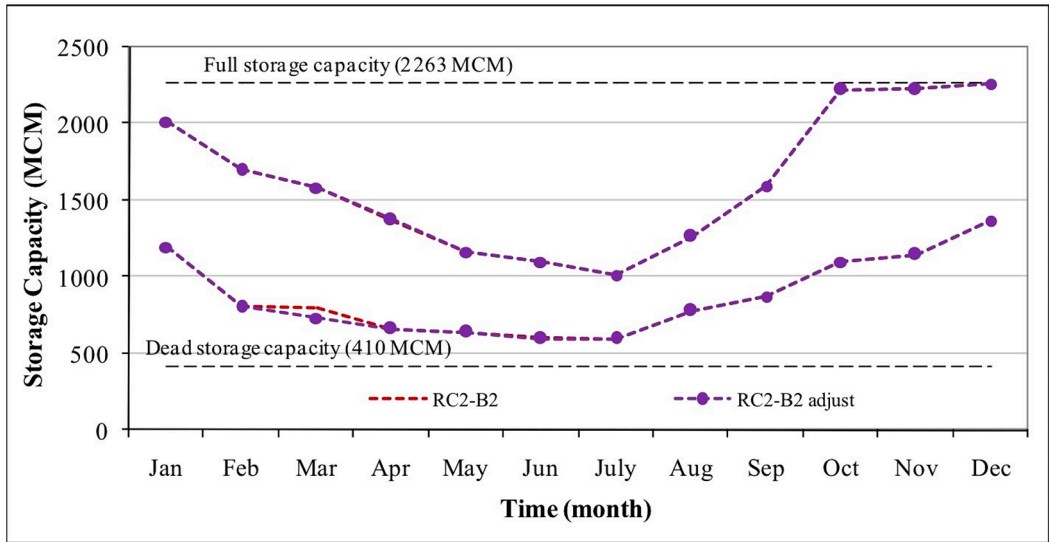

**Figure 20.** Optimal rule curves of the Ubolratana Reservoir (B2 Scenario) adjusted after stakeholder participation [9].

## 6. Conclusions

Reservoir rule curves are necessary guidelines for controlling the storage and release of water from a reservoir. Generally, the rule curves need to be improved when the data of water flowing into the reservoir and downstream water demands change. The inflow into the reservoir is affected by land use in the upstream area and climate changes. Generally, this inflow is estimated by using the SWAT model, which considers both land use and climate changes. The popular land use change model is CA Markov. The climate models are PRECIS and GCM.

Many optimization techniques have been applied alongside the reservoir simulation model in order to improve the reservoir rule curves. They introduce progressively simple to complex techniques, using the trial and error technique with the reservoir simulation model, dynamic programing, heuristic algorithm, and swarm-intelligence-based and evolutionary algorithms. The decision variables are the optimal rule curves that are defined as the upper rule and lower rule curves. There are 24 decision variables for the monthly rule curves of a single reservoir and they should be multiplied for a multi-reservoir system, consisting of

12 values from the upper rule curves and 12 values from lower rule curves. The best values of these decision variables are archived by optimization techniques integrated with the reservoir simulation models using smoothing function constraints. The suitable future rule curves for future condition situations are the obtained rule curves from the optimization technique integrated with the reservoir simulation model considering future conditions. Furthermore, the future inflow from upstream into reservoir should be used in the reservoir simulation model with up-to-date downstream water demands. The estimated future inflow into the reservoir is required to assess land use and climate changes as well as stakeholder participation. In future studies, short-term operation and weekly rule curves should be investigated to increase their use in reservoir operations.

**Author Contributions:** Conceptualization, A.K.; methodology, A.K.; validation, A.K.; formal analysis, A.K.; investigation, A.K.; writing—original draft preparation, R.T., R.N. and A.K.; writing—review and editing, H.P., K.S., S.M.A., R.H., R.T., R.N. and A.K.; supervision, H.P., K.S., S.M.A., R.H., R.T., R.N. and A.K. All authors have read and agreed to the published version of the manuscript.

**Funding:** This work was financially supported by Mahasarakham University.

**Data Availability Statement:** This study did not report any data.

**Conflicts of Interest:** The authors declare no conflict of interest.

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
