# Peer review of "Application of Optimization Techniques for Searching Optimal Reservoir Rule Curves: A Review"

_water, doi:10.3390/w15091669_

Round 1
Reviewer 1 Report
(1) It seems that the Figure 5 cannot illustrate what the authors want to convey. Please revise the figure or associate statements.
(2) Line 77-79: I suggest authors give some conceptual diagrams of the operating rules used for water supply, power generation and flood control, and clarify their advantages when compared with HR or SOP.
(3) Line 137-140: Could you give some references to support your statements? Because some objectives are rarely found in the literature.
(4) To avoid fluctuation of the rule curves, optimizing the key points of the rule curves can be an effectively way. So the authors could add some materials on this issue.
(5) Operation of multi-reservoir system is more complicated. In this context, how to derive operating rule curves? Please give some comments and discussions on this issue.
Author Response
Thank you very much for your comments and suggestions. We really appreciate the reviewers' comments, which are very detailed and very helpful in improving our manuscript. We have made a minor revision to our manuscript. We have improved our manuscript in the following aspect: (1) we have rewritten the abstract to describe the maim finding of this paper; (2) we have rewritten the introduction to better introduce the main justification and the objective of this paper; (3) we have rewritten the conclusion to address the future research possibilities that merit further investigation of the work; (4) we have rewritten the whole paper according the comments and suggestions; (5) we have added some relevant review work as suggested by the reviewers; (6) we have rechecked the whole manuscript and have corrected some mistakes.
Reviewer 2 Report
The goal of the study is lacking novelty.
The field applicability of the work and shortcoming that you observed
from the literature should be clearly explain.
What could be the future research possibilities that merit further
investigation of the work should be explained clearly.
The conclusion part should be specific, needs to be restructured.
The Plagiarism of the manuscript should be less than 10% right now it is 38%.
Otherwise the writing style of the manuscript was good.
Author Response

(The authors gave the same response as above.)

Reviewer 3 Report
The paper is a review article on applications of optimization techniques connected with reservoir simulation models in searching for optimal rule curves. The authors have reviewed over 180 reference papers, which makes this paper a good topic selection, strong practicability and sufficient literature research.
However, some of the technologies in the paper are not clearly stated and required more explicit description.
1. In the chapter "4.3.2 The Shuffled Frog Leaping Algorithm", there is no example of its application. It is suggested to add the application case of this algorithm, and show the effect. In addition, I suggest you to add quantifiable data to the effect of application cases of other algorithms to prove the practicability of these algorithms.
2. In the chapter "4.5 Swarm Algorithms", you only introduce the bionics principle of a part of the algorithms (such as "4.5.2. Cuckoo Search", "4.5.4. Firefly Algorithm" and "4.5.6. Gray Wolf Optimizer"), but did not explain the operation logic and use method of these algorithms themselves, which is easy to make some readers difficult to understand. A brief and direct description of this algorithm is recommended.
Author Response

(The authors gave the same response as above.)

Reviewer 4 Report
Please see attached

Round 2
Reviewer 2 Report
Dear Authors
The present version of manuscript is technically accepted but you can not ignore the plagiarism. Still the plagiarism of the current version of manuscript is around 36% (Find in attachment) which is not in the journal ethics. So please reduce the Plagiarism less than 15- 20%, which is acceptable.
Thank you
